# NAA40 and NAC cooperate in co-translational histone acetylation in humans

Dandan Guan [1,3], Timo Denk [1,3], Ariel Klavaris[2], Matthias Thoms[1], Otto Berninghausen [1], Birgitta Beatrix [1], Antonis Kirmizis [2] & Roland Beckmann [1] ✉

N-terminal acetylation is an abundant and predominantly co-translational modification in eukaryotes that profoundly affects folding, compartmentalization fidelity and turnover of target proteins. Unlike other N-acetyltransferases, human NatD is composed solely of the catalytic subunit NAA40 and exclusively modifies histone proteins H2A and H4. However, the molecular details of co-translational NAA40 activity have remained elusive. Here, we show biochemically and by cryo-EM how NAA40 activity is coordinated at the ribosomal peptide tunnel exit involving the NAC complex. We demonstrate that the NAA40-NAC interaction is required for efficient ribosome binding and histone acetylation. Furthermore, we provide insights on the potential coordination of methionine removal and subsequent NAA40-mediated acetylation by formation of a multienzyme complex on the ribosome involving METAP1. Therefore, our results illustrate the details of N-terminal histone acetylation by NAA40 and highlight the role of NAC as a general coordinator of nascent protein modification.

Approximately 60% of all yeast and 80% of all human proteins are N-terminally acetylated (Nt-acetylation)[1], a modification which in most cases is added co-translationally, when the nascent polypeptide chain emerges from the ribosome. For this modification, N-terminal α-acetyltransferases (NATs) catalyze the transfer of an acetyl group from acetyl-CoA to the α-amino group of the nascent peptide chain, often requiring the removal of the N-terminal methionine. Of the eight distinct NATs that have been identified in eukaryotes (NatA-NatH), five act co-translationally on the ribosome (NatA to NatE)[2]. At the ribosome, NATs compete with each other and an array of other enzymes, which also bind to the ribosomal tunnel exit to ensure the proper maturation of the newly made polypeptide[3]. Protein Nt-acetylation has been shown to be decisive for a diversity of general processes in the cell, such as protein-protein interactions, subcellular localization, protein targeting and protein half-life. Yet, also more specific activities such as sister chromatid cohesion and chromosome condensation in the nucleus are dependent on accurate Nt-acetylation[2]. Of the five NATs acting co-translationally, four (NatA, NatB, NatC, NatE) are responsible for the vast majority of protein Nt-acetylation and typically consist of a catalytic subunit and one or two accessory subunits that mediate the interaction with the ribosome[4–7]. NAA40 (or NatD), on the other hand, is one of the most specialized NATs since in humans it Nt-acetylates exclusively histone proteins H2A, H4 and histone variant H2A.X[8]. Due to the distinct structural features of its catalytic domain, NAA40 provides specificity for the N-terminal four amino acids of its cognate substrates[9,10]. In contrast, NATs with a broad substrate range, like NatA and NatE, mainly recognize the first two target residues[11,12]. Furthermore, NAA40 binds directly to the ribosome, therefore lacking any accessory subunits[9].

NAA40 is active only after the starter methionine has been removed by the methionine amino peptidases METAP1 or METAP2, thereby exposing the recognition sequence Ser-Gly-Arg-Gly[10].

[1]Gene Center, Ludwig-Maximilians-Universität München, Munich, Germany. [2]Department of Biological Sciences, University of Cyprus, Nicosia, Cyprus. [3]These authors contributed equally: Dandan Guan, Timo Denk. ✉e-mail: beckmann@genzentrum.lmu.de

Importantly, Nt-acetylation of these histone proteins antagonizes other histone modifications, specifically H2A/H4S1Ph (Ph, phosphorylation) and H4R3me (me, methylation), thereby influencing chromatin activity and gene regulation[13,14]. Most notably, NAA40 levels are altered in various cancers, and NAA40 has been shown to promote oncogenesis (summarized in ref. [13]), which is why NAA40 has been recognized as a potential therapeutic drug target. However, it is not known how NAA40 interacts with the ribosome to modify nascent H2A and H4.

Here we present the structure of human NAA40 bound to the 80S ribosome. We discovered that NAA40 is positioned near the ribosomal tunnel exit and that its ribosome interaction largely depends on the nascent polypeptide-associated complex (NAC). Notably, we demonstrate that within cells, the ubiquitin-associated (UBA) domain of NAC is required for a productive interaction of NAA40 with the ribosome. Finally, we provide structural evidence for the coordination of

methionine excision and NAA40-mediated Nt-acetylation by formation of a multienzyme complex on the ribosome.

## Results

### NAA40 and NAC co-associate with ribosomes in vivo

The N-terminal histone acetylation activity of NAA40 occurs both post- and co-translationally, however, the significance of the co-translational activity has been poorly characterized[9,15,16]. To better understand this co-translational activity of NAA40, we first aimed at investigating the association of NAA40 with ribosomes in vivo. Therefore, we expressed NAA40 with a C-terminal 3C protease cleavage site and 3xFLAG tag in HEK293T cells. We isolated the tagged NAA40 under native conditions from cell lysate and indeed enriched 80S ribosomes (Fig. 1a). In addition to ribosomes, we noticed enrichment of histones as well as XRCC5/6 and PRKDC (Fig. 1a), which are essential actors in DNA damage repair via nonhomologous end joining[17,18]. Association with

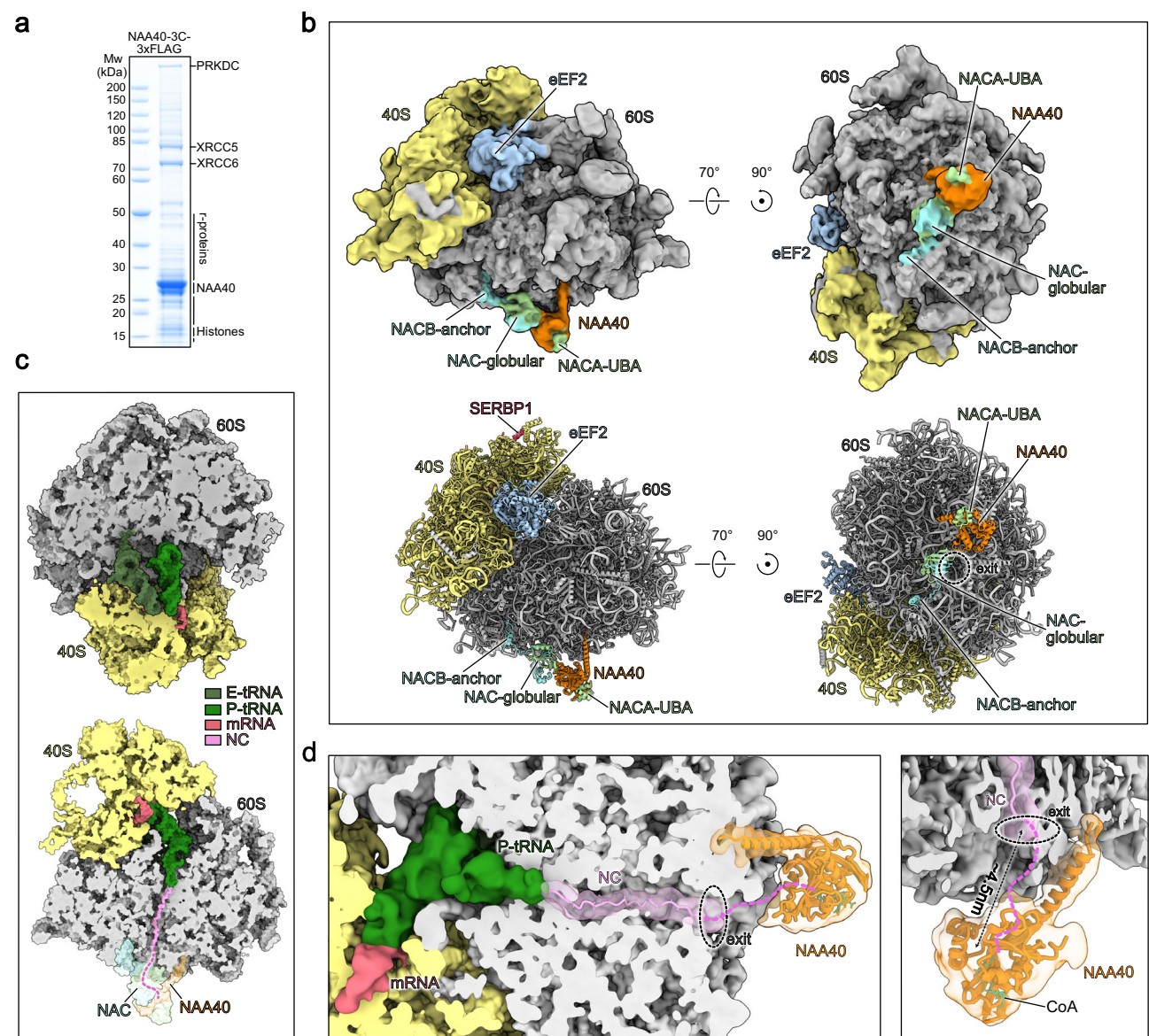

**Fig. 1 | NAA40 and NAC co-assemble on the ribosome in vivo. a** SDS-PAGE gel stained with Coomassie of the elution fraction of tagged NAA40 purified under native conditions. Prominent bands were annotated based on mass spectrometry results (see Methods). **b** Cryo-EM density maps (top) and molecular models (bottom) of the combined hibernating ribosomes with NAA40 and NAC bound. **c** Cut-through view of surface representations of actively translating 80S bound with

NAA40 and NAC. The translational state (top) or the nascent polypeptide chain is highlighted (bottom). **d** Close up view of cut throughs of the surface representations of actively translating 80S, focusing on the nascent chains (NC; left) or the distance of NAA40 to the peptide exit tunnel (right). NAC was omitted for clarity. Source data are provided as a Source data file.

histones was expected, as H2A, H4, and H2A.X are established NAA40 substrates[8] and the enzyme localizes to both cytosol and nucleus[9]. Also, the enrichment of DNA repair factors is in line with the recently reported involvement of the NAA40 yeast homolog in DNA damage response[16].

To elucidate the molecular details of the NAA40-ribosome interaction, we subjected the isolated NAA40 complexes to cryo-EM single particle analysis (SPA). The majority of 2D classes obtained were representative of 80S ribosomes, and only a few classes corresponded to co-purified nucleosomes (Supplementary Fig. 1). After 3D classification of ribosomal particles, we obtained classes representing translating ribosomes but mainly inactive/hibernating ribosomes, likely as a result of NAA40 overexpression. Yet, all these classes shared a distinct density on the large ribosomal subunit at the peptide exit site and thus were combined to gain an improved overall resolution (2.7 Å) (Supplementary Fig. 2a–d upper panels, Supplementary Table 1). The resulting combined reconstruction mainly resembled a hibernating state with SERBP1, eEF2 and E-site tRNA bound (Fig. 1b)[19,20] and allowed us to identify NAA40 and, unexpectedly, NAC binding together in the vicinity of the peptide exit. NAC is a heterodimeric complex consisting of NACA and NACB that has been recognized as a constituent component of the translation machinery, coordinating co-translational modifications and protein translocation as well as acting as an early chaperon on the ribosome[3,21,22]. To assess potential interactions between NAC and NAA40, we performed an AF2 (AlphaFold 2) multimer analysis[23,24], which confidently predicted an interaction of the NACA UBA-domain and the catalytic domain of NAA40 (Supplementary Fig. 3a–c). The interaction was predicted to be mediated via both negatively charged (NACA: E177, D180, D205; NAA40: R50, K153, R216) and hydrophobic residues (NACA: L183, I210; NAA40: F45, L161) of the UBA domain (Supplementary Fig. 3d), and was in agreement with additional density observed on NAA40 in our reconstruction (Fig. 1b). To gain further insights into the co-translational substrate engagement of NAA40 we isolated a smaller but defined class of particles representing actively translating ribosomes with both P- and E-site tRNAs (Fig. 1c, Supplementary Figs. 1 and 3e, f). Here, NAA40 and NAC associate with the ribosome in essentially the same manner as for inactive ribosomes. Despite lower overall resolution (2.9 Å) (Supplementary Fig. 2a–d lower panels, Supplementary Table 1) the nascent polypeptide chain could still be traced from the P-site tRNA to the peptide tunnel exit site but density is lost after exiting the ribosome and could not be unambiguously assigned within the catalytic domain of NAA40 (Fig. 1c, d left panel and Supplementary Fig. 3e, f). However, using a crystal structure of substrate analogue-bound NAA40[25] (Supplementary Fig. 3g) as a reference point, one can estimate that nascent chains need to span a distance of about 4.5 nm or at least 12 amino acids after leaving the ribosome to reach the active site of NAA40 (Fig. 1d right panel). Compared to recent structures of the ribosome, NAA40/NatD is similarly positioned to NatA/E acetyltransferase complexes[6,7] (Supplementary Fig. 3h, i), but nascent polypeptides theoretically require at least 16 or 20 amino acids outside the peptide exit to reach their respective catalytic subunits NAA10 and NAA50 (Supplementary Fig. 3j).

## Molecular details of the NAA40-NAC ribosome interaction

Next, we analyzed the structural basis of the interaction of NAA40 and NAC with the ribosome. When looking at the overall positioning of NAA40 and NAC on our best resolved ribosome (2.7 Å, Supplementary Fig. 2a–d), both proteins are situated in close proximity to the peptide exit site for immediate engagement with emerging substrate polypeptides (Fig. 2a). Notably, NAA40 consists of two parts: the catalytic GCN5-related N-acetyltransferase (GNAT) domain and an N-terminal helical extension (α0) that is unique to NatD compared to other NATs[10]. The catalytic domain resides on top of ribosomal protein uL24 and the 5.8S rRNA helix H7 with its substrate binding cavity facing

towards the peptide tunnel exit (Fig. 2a, b left, Supplementary Fig. 4a). However, the lower local resolution of the GNAT domain (Supplementary Fig. 4a) as well as the minimal contact points with uL24 and 5.8S H7 indicate a certain degree of flexibility for the catalytic domain which might hint at a dynamic conformational adaptability for its continuously elongating substrates. The N-terminal α0 helix on the other hand is well resolved (Supplementary Fig 4c), anchoring NAA40 to the ribosome via interactions with uL24 as well as rRNA helices H19, H24 and H46 (Fig. 2b right, Supplementary Fig. 4b). Its highly positively charged portion engages with the negatively charged surface of rRNA elements H19 and H24 (Fig. 2c left and middle, Supplementary Fig. 4c and d). The unstructured very N-terminus of NAA40 further protrudes towards H46, held in place by interaction of lysine K4 with H19 and π-charge interaction of arginine R3 with guanine G409 of H24 (Fig. 2c right), and is essentially buried at the convergence site of the three rRNA segments. In essence, the α0 helix of NAA40 appears to act as a substitute for additional auxiliary subunits, which have been shown to serve as ribosome tethers for other NATs[4–7]. Furthermore, NAA40 is highly conserved in vertebrates for both its GNAT domain and N-terminus (Supplementary Fig. 5a, b), supporting the notion of functional importance of the α0 helical element, including its ribosome interaction capabilities.

The NAC complex interacts with the ribosome as observed before[26]. It is anchored to the ribosome via the NACB N-terminus, which is attached to ribosomal protein eL22 (Fig. 2d, Supplementary Fig. 4e, f). The NAC globular domain is positioned near the peptide exit site and close to an unstructured loop of the NAA40 catalytic domain (191–213aa) (Fig. 2e). However, similar to the NAA40 GNAT domain, the lower local resolution (Supplementary Fig. 4e, f) suggests a level of positional flexibility for the NAC globular domain possibly allowing it to adapt to the nascent polypeptide. Consistent with this assumption, we observed the NAC beta sandwich positioned closer to the peptide exit in our reconstruction of actively translating ribosomes than for hibernating 80S (Fig. 2f). Lastly, the NACA UBA-domain was found interacting with the ribosome distal side of the NAA40 catalytic domain, as predicted by AF2 (Fig. 2g, Supplementary Fig. 3a–d). This suggests that, similar to the N-terminal acetyltransferase complexes NatA/E, which require the NACA UBA interaction for stable ribosome association[6], NAA40 ribosome binding might also depend on NAC. Further, when comparing NAA40 with NatA/E, it becomes evident that they share a common binding site as the NatA/E auxiliary subunit NAA15 also binds to rRNA helices H19 and H24[6,7] (Supplementary Fig. 4g and h). Taken together, this architecture implies a similar mode of interaction of NAA40 and NatA/E with NAC. Yet, it also highlights the competitive environment of the peptide exit site for co-translational modification factors that requires appropriate coordination for each target peptide.

## NAC coordinates the co-translational activity of NAA40

Recently, NAC has been identified as a master regulator for the recruitment of co-translational protein modification enzymes such as METAP1[26], NatA/E[6,7] and for protein translocation complexes like SRP to the ribosome[27]. The consistent coappearance of NAC and NAA40 in our ex vivo cryo-EM data, as well as a similar interaction with the NACA UBA-domain for NatA/E suggest comparable recruitment of NAA40 to translating 80S by NAC. To test this hypothesis, we performed in vitro binding assays with purified recombinant NAA40-FLAG together with NAC and NAC-ΔUBA complexes to assess their association with purified 80S ribosomes (Supplementary Fig. 6a–c). The respective components were incubated together with anti-FLAG resin, immobilized via tagged NAA40 and bound complexes were eluted with FLAG peptide. NAA40 alone bound only weakly to ribosomes, but in the presence of NAC, enrichment of ribosomes was strongly increased (Fig. 3a, Supplementary Fig. 6d). Conversely, when incubated with NAC-ΔUBA, the ribosome association of NAA40 was lost entirely

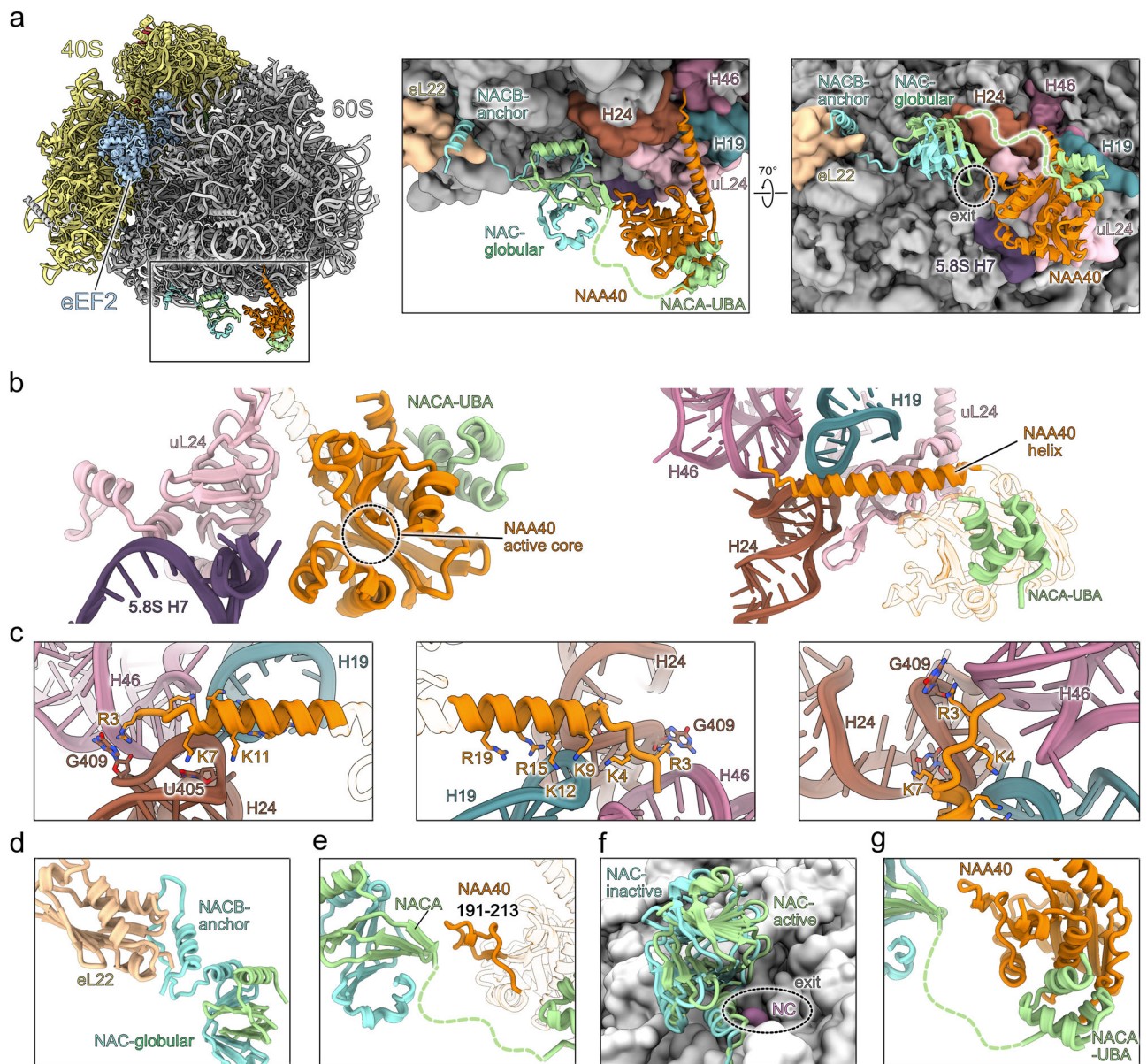

**Fig. 2 | Molecular details of the NAA40-NAC ribosome interaction. a** Overview of the NAA40-NAC binding site at the peptide exit site. **b** Close ups of the interaction sites of the GNAT domain (left) and the N-terminal α0 helix (right) of NAA40 with the ribosome. **c** Details of the interactions of the positively charged residues of the NAA40 α0 helix and the unstructured N-terminus with rRNA helices H19, H24 and H46. **d–g** Close ups of the interactions of NAC with NAA40 and the ribosome. **d** NACB anchor-eL22 interaction. **e** Contacts of NAC beta-sandwich and NAA40 GNAT domain. **f** Comparison of the NAC globular domain positioning between active (green) and hibernating 80S (cyan). **g** NAA40-NACA-UBA domain.

(Fig. 3a). To further examine the negative effect of NAC-ΔUBA on NAA40 ribosome association, we performed another binding assay but this time titrated NAC against NAC-ΔUBA (Supplementary Fig. 6e, f). Here, increasing the proportion of NAC-ΔUBA led to a decrease in associated ribosomes until binding is again essentially lost when only NAC-ΔUBA is added. We speculate that the high affinity of NAC for the ribosome, in combination with its flexible positioning around the peptide exit site, might lead to a steric block for NAA40 binding in the case of NAC-ΔUBA. Therefore, it seems that the UBA-domain acts in an almost allosteric manner, both assisting and allowing proper NAA40 ribosome binding. Taken together, NAA40 binding to ribosomes appeared indeed largely dependent on NAC in vitro and, similar to NatA/E and in agreement with our structure, the NAC interaction with NAA40 is mediated by the NACA UBA-domain.

Based on our structural data, we identified the N-terminal α0 helix of NAA40 as its main anchor for ribosome association (Fig. 2b, c,

Supplementary Fig. 4c, d). Recently, a second isoform of NAA40 has been reported that lacks the first 21 amino acids (NAA40S) generated either by alternative translation initiation at M22 of the canonical NAA40 transcript or expression of a shorter transcript variant[28,29]. Interestingly, this part of the α0 helix contains the entirety of highly conserved, positively charged residues responsible for interaction with rRNA elements (Fig. 3b, compare Fig. 2c). In order to assess the contribution of the positively charged N-terminus to ribosome association, we performed our binding assay also with recombinantly purified short form, NAA40S-FLAG. As expected, we could not observe any binding to ribosomes of NAA40S, whether on its own or in the presence of NAC (Fig. 3a last two lanes, Supplementary Fig. 6d). This demonstrates that the positively charged N-terminus is essential for NAA40 ribosome binding and also suggests that the NAA40S isoform is unlikely to participate in co-translational Nt-acetylation, but possibly active in post-translational modification in the cytosol. Furthermore,

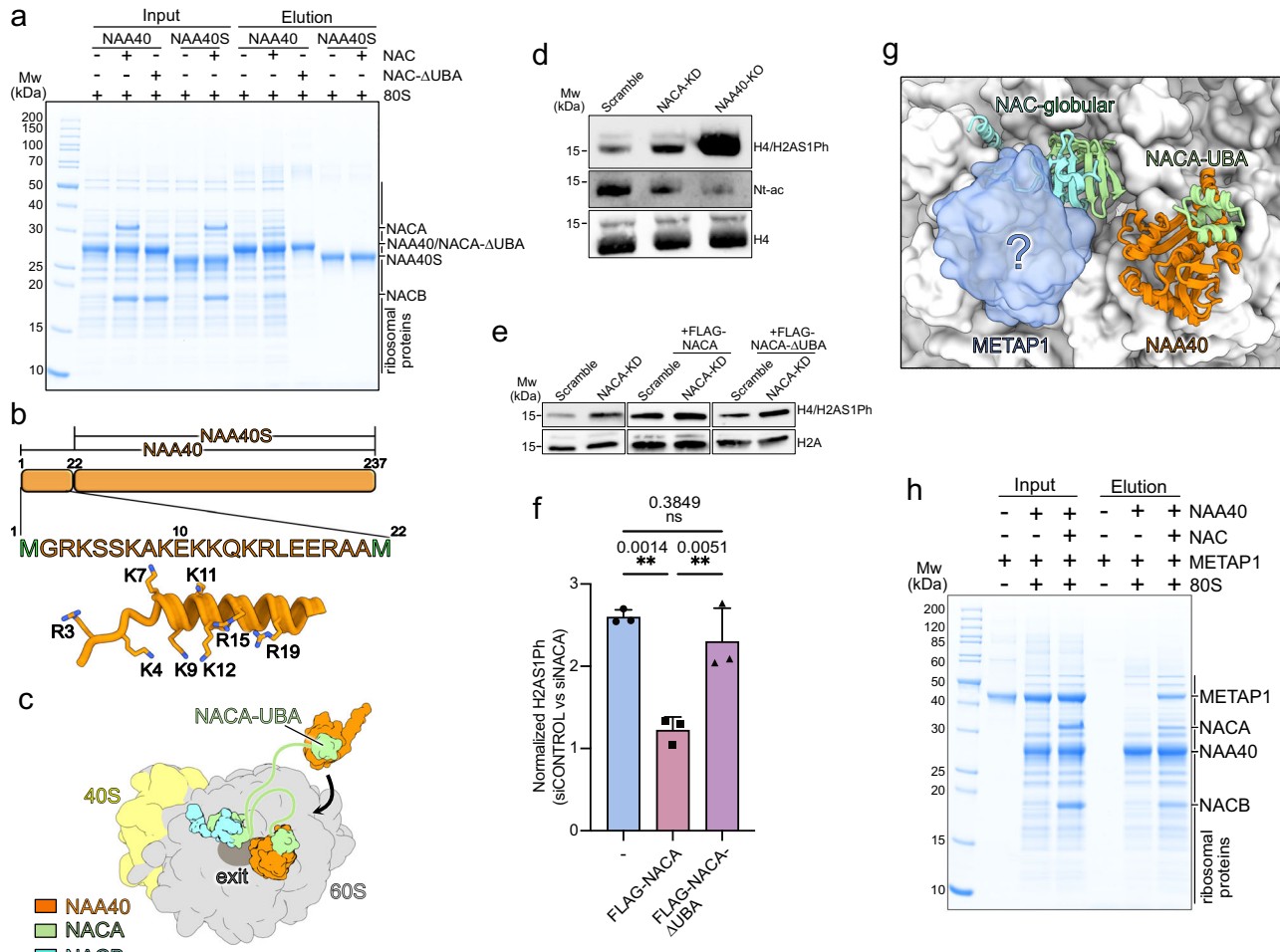

**Fig. 3 | NAC facilitates co-translational acetylation by NAA40 and coordination with METAP1. a** In vitro binding assay of NAA40 and NAA40S with NAC and 80S. SDS-PAGE gel of input and elution fractions stained with Coomassie. **b** Schematic comparison between NAA40 and NAA40S (top). Molecular model of the first 21 amino acids exclusive to full-length NAA40 with positively charged amino acids interacting with rRNA shown (bottom). **c** Cartoon representation showing the recruitment of NAA40 to the ribosome by NAC via the NACA-UBA domain. **d** Western blot analysis of histone extracts from HCT116 Scramble, NACA-KD and NAA40 knock-out (KO) cells, using antibodies against N-terminal acetylation (Nt-Ac), its antagonistic mark H4/H2AS1Ph and H4 as a loading control. **e** Western blot analysis of histone extracts from HCT116 scramble, NACA-KD transduced with FLAG-NACA (WT) or FLAG-NACA (ΔUBA) siRNA resistant plasmids, using antibody

against NAA40 antagonistic mark H4/H2AS1ph and H2A as a loading control. **f** Quantification of western blot signals H4/H2AS1Ph relative to H2A from (**e**). All statistical analyses were performed using unpaired One-Way ANOVA (ns no significance, **$P < 0.01$) (bottom), $n = 3$ biological replicates/group. $P$ values up to 5 digits are shown. Error bars represent standard error of mean (SEM). **g** Overlay of ribosome bound NAA40-NAC model with ribosome bound METAP1 (PDB: 8P2K), showing no steric clashes. The question mark indicates that based on the overlay the potential for co-assembly of METAP1 with NAA40-NAC on the ribosome is given and tested in (**h**). **h** In vitro binding assay of NAA40 and METAP1 with NAC and ribosomes. SDS-PAGE gels of input and elution fractions stained with Coomassie. Source data are provided as a Source data file.

we conclude that the interaction of NAC with NAA40 via the NACA UBA-domain is necessary (Fig. 3c) but not sufficient to mediate a stable ribosome association.

Next, we aimed to determine whether NAC had an effect on NAA40-mediated histone Nt-acetylation within cells. Therefore, we raised an antibody recognizing Nt-Ac on an SGRG sequence shared by H2A and H4, but with the antibody detecting primarily histone H4 Nt-acetylation (Supplementary Fig. 6g, h). Using this antibody, we probed the acetylation as well as the antagonistic H4/H2AS1Ph phosphorylation levels in NACA knockdown (Supplementary Fig. 6i) and NAA40 knockout (NAA40-KO) cell lysates (Fig. 3d). As expected, in NAA40-KO cells the signal for H4 Nt-acetylation was significantly reduced but not completely abolished, probably due to weak H4 backbone recognition by the antibody (Supplementary Fig. 6h). At the same time, the H4/H2AS1Ph signal was strongly increased compared to wild type lysate. More interestingly, knockdown of NACA also led to a decrease in H4 Nt-acetylation and an increase in H4/H2AS1Ph levels (Fig. 3d). As

expected, the effect was not as profound as for NAA40-KO cells, likely due to the incomplete dissociation of NAA40 from ribosomes in the absence of NAC and/or due to the remaining post-translational modification activity of NAA40. Additionally, while the almost complete reduction in NACA mRNA levels confirms the efficient NACA knockdown (Supplementary Fig. 6i), we cannot exclude that residual H4 Nt-acetylation may be present due to remaining NACA protein. Importantly, NACA depletion did not affect NAA40 expression levels or its localization (Supplementary Fig. 6i and j, respectively), suggesting that its effect on histone Nt-acetylation is primarily due to the weaker association of NAA40 with the ribosome. To further determine whether the observed decrease in histone Nt-acetylation is directly dependent on the NACA UBA-domain interaction we complemented cells that were depleted of NACA with siRNA resistant constructs expressing either N-terminally FLAG tagged NACA or NACA-ΔUBA (Supplementary Fig. 6k) and monitored the antagonistic H4/H2AS1Ph levels as a proxy for Nt-acetylation (Fig. 3e). While the phosphorylation

levels were increased three-fold when depleting NACA compared to non-targeting siRNA, as observed above (Fig. 3e, f), the levels of H4/H2AS1Ph were equivalent between control cells and cells depleted for NACA but complemented with exogenous full-length NACA (Fig. 3f). However, when NACA-depleted cells were complemented with NACA-ΔUBA the H4/H2AS1Ph levels remained higher again by more than two-fold compared to the control cells, indicative of compromised Nt-acetylation activity (Fig. 3f). Taken together, our in vitro and in vivo data indicate a substantial role of the NACA UBA-domain interaction with NAA40 for ribosome binding/recruitment (Fig. 3c) and thereby for overall histone Nt-acetylation in cells.

Another important aspect of H2A and H4 Nt-acetylation is the preceding removal of the initiator methionine by methionine amino-peptidases (MetAPs) to expose the target serine and how this MetAP activity is coordinated with respect to NAA40 ribosome binding and activity. As recently shown in vitro, METAP1, NAC, and the acetyl-transferase complex NatA/E can assemble at the peptide exit site simultaneously to presumably facilitate a seamless substrate handover between METAP1 and NatA/E[6,7]. Since NatA/E rely on NAC in a similar fashion as NAA40[26] we next examined whether NAC could also coordinate the concomitant recruitment of METAP1 and NAA40. When superimposing our ex vivo NAC-NAA40 structure with ribosome-bound METAP1, no steric clashes were observed (Fig. 3g). Therefore, we performed another in vitro binding assay with NAA40-FLAG, 80S and METAP1 with or without NAC. As shown above, omission of NAC alone showed weak association of NAA40 with ribosomes (compare Fig. 3a), and notably also resulted in no co-enrichment of METAP1[26] (Fig. 3h and Supplementary Fig. 6l). However, inclusion of NAC resulted in enhanced NAA40 ribosome association and also binding of METAP1 (Fig. 3h), confirming the potential for a multienzyme assembly on the ribosome coordinated in the presence of NAC.

### NAA40 and METAP1 can form a multienzyme assembly on the ribosome in vitro

To gain detailed molecular insights into the formation of a ternary complex consisting of NAA40, NAC and METAP1 on the ribosome, we in vitro reconstituted these components with purified 80S ribosomes and performed cryo-EM SPA. After extensive 3D classification (Supplementary Figs. 7 and 8a–d), we observed a class of particles that contained only NAA40 together with NAC, which essentially resembled our ex vivo cryo-EM data with respect to positioning of the factors and binding of the NACA UBA-domain to NAA40 (Supplementary Fig. 9a–d).

More interestingly, however, we isolated two classes of complexes with NAA40-NAC bound with additional density at the peptide exit tunnel site representing two different conformational states of METAP1 (State1 and State2) (Fig. 4a, b, Supplementary Figs. 7, 8, 10a, b). For both states, METAP1 was binding to the groove between ribosomal protein uL23 and rRNA helix H59 with the active site pointing towards the peptide exit site, as reported before[6,7,26] (Fig. 4c, Supplementary Fig. 10c, d). While the NACA UBA-domain on NAA40 was again clearly visible, density for the flexible zinc finger domain of METAP1 and the unstructured NACB C-terminus that contains the hydrophobic METAP1 interaction motif (NACB isoform 1: 190-VPDLV-194, isoform 2: 146–150) (Fig. 4d, e, Supplementary Figs. 10a, b and 10f–h) was not observed, as described previously[26].

When comparing the two conformations of METAP1, the enzyme was tilted closer towards the NAC globular domain and the peptide exit in State1 than in State2 (Fig. 4f). This slight conformational variability entails discrete differences for nascent chain length requirements for the enzymatic activity of METAP1. Therefore, the emerging peptide needs to span a distance of about 4.4 or 5.1 nm for State1 or State2, respectively (Fig. 4f). This translates to at least 12 or 14 amino acids that must have protruded from the ribosome exit and is in the same distance range as for NAA40 (12+ amino acids, compare Fig. 1d).

The almost identical distances required to reach the active sites, as well as the dependency on the two different subunits of the NAC complex for NAA40 (NACA) and METAP1 (NACB), hint towards both enzymes being recruited simultaneously on the ribosome to process the nascent polypeptide chain and to potentially achieve rapid substrate exchange (Fig. 4g). This would essentially result in a similar situation as suggested for NatA/E which displayed co-association on the ribosome with NAC and METAP1[6,7]. However, while METAP1 requires NAC for sufficient binding, METAP2 does not[26,30], even more so since its additional ribosome interaction domain that it shares with, e.g. EBP1, renders ribosome binding mutually exclusive with NAC[7]. Yet, both MetAPs show a degree of functional redundancy in eukaryotes[31–33]. Consistent with their overlapping functions but contrary to the contribution of NAC to NatA/E ribosome binding, a complex of METAP2-NatA-80S has recently been reconstituted in vitro and structurally characterized[7] (Supplementary Fig. 10i, j). For NAA40, on the other hand, simultaneous ribosome association with METAP2 is not possible as they would sterically clash (Supplementary Fig. 10k). Taken together, our structural data highlight the potential of NAC to orchestrate the coupling of methionine excision, specifically by METAP1, with Nt-acetylation by NAA40 on the ribosome. However, similar to previous reports on METAP1/2 and NatA/E, the data are largely in vitro based, and a sequential order of 80S binding and nascent chain modification events in vivo cannot be excluded for NAA40. Such a sequential order of interaction (Fig. 5) would be in agreement with our observation that we could not detect classes of ribosomes with NAA40, NAC and METAP1 bound concomitantly in our ex vivo preparations.

## Discussion

Here, we shed light on the co-translational activity of human NatD/NAA40 and its engagement with the ribosome using ex vivo cryo-EM snapshots. Thereby, we identified the ubiquitous NAC complex as a cofactor of ribosome-bound NAA40. Furthermore, we show that in vitro the interaction between NAA40 and the NACA UBA-domain is required for sufficient ribosome binding, and conversely, its loss leads to deficient H2A and H4 Nt-acetylation in vivo. This demonstrates the important role of NAC in the orchestration of co-translational events at the ribosomal peptide exit tunnel, as observed for other factors before[6,7,26,27]. However, since NAC can recruit NatD to the ribosome in the same fashion as NatA/E or SRP, it still remains intriguing how differentiated substrate selection is coordinated in detail for these competing factors and how modification fidelity is ultimately achieved. Unraveling this selection process is not only of importance because of the overlapping interactions with NAC but also their coinciding binding sites on the ribosome. In addition, all of these factors can associate at the ribosome independent of the presence of the nascent poly-peptide chain in vitro and possibly also in vivo. Therefore, we can only speculate whether, in cells, competitive binding and continuous exchange of modifying enzymes at the peptide exit site occur until the correct target substrate is available or whether a pre-recognition by NAC is relayed into matching enzyme recruitment.

Despite the lack of any density in our ex vivo structures for METAP1 (or METAP2), we provide structural evidence in vitro for methionine excision by METAP1 and subsequent Nt-acetylation by NAA40 to potentially proceed in a concerted manner. NAC may coordinate the concomitant instead of sequential binding of both enzymes to the ribosome by recruiting NAA40 via the NACA UBA-domain and METAP1 via the NACB C-terminus (Fig. 5). This enzyme assembly would allow for seamless substrate exchange which is, based on our structural data, in agreement with the similar distances for the nascent polypeptide chain to reach the two enzymes for processing. Very similar data have been presented in a recent publication by the Ban lab[34], also suggesting concomitant binding of METAP1 and NAA40 mediated by NAC in vitro. These data are in line with previous structures of NatA/E and METAP1 or METAP2 being simultaneously

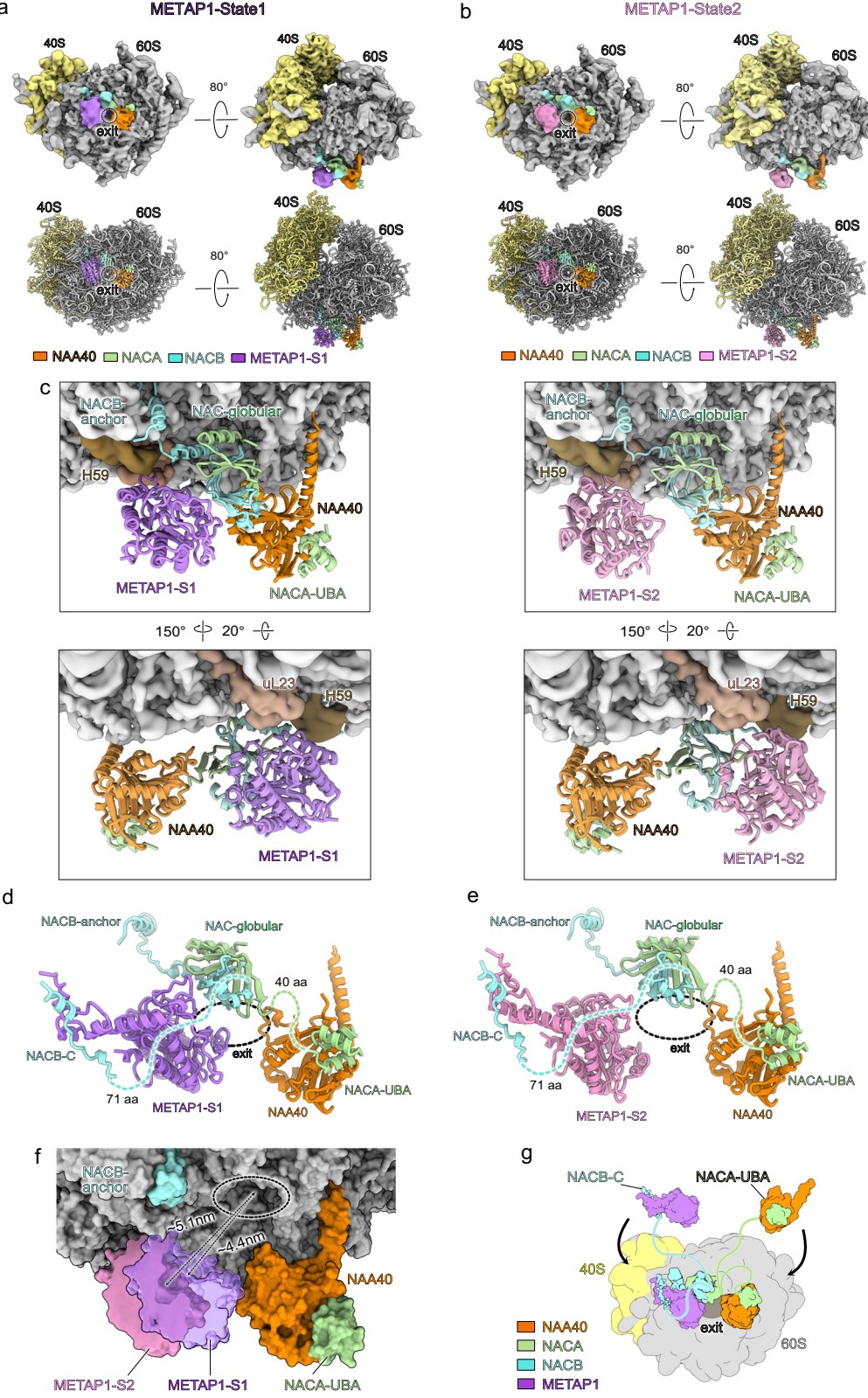

**Fig. 4 | METAP1 and NAA40 form a multienzyme assembly on the ribosome in vitro.** Local resolution filtered cryo-EM maps (top) and molecular models (bottom) of **a** METAP1-State1 and **b** State2. **c** Overview of the NAA40-METAP1-NAC ribosome binding site for State1 (top, METAP1-S1) and State2 (bottom, METAP1-S2). Molecular models of NAA40, METAP1 and NAC as assembled around the peptide exit for **d** State1 and **e** State2. Molecular models for METAP1 were substituted with AF3 predictions for the METAP1 zinc finger−NACB C-terminus interaction that is

not visible in our structure. Flexible connection between the NACB C-terminus and the NACA-UBA domain with the NAC globular domain is indicated.
**f** Overlay of METAP1 State1 and State2 indicating the distances between the METAP1 active sites and the peptide exit site. Molecular models are shown as surface representations, and the NAC globular domain was omitted for clarity.
**g** Cartoon representation illustrating the simultaneous recruitment of NAA40 and METAP1 via NAC.

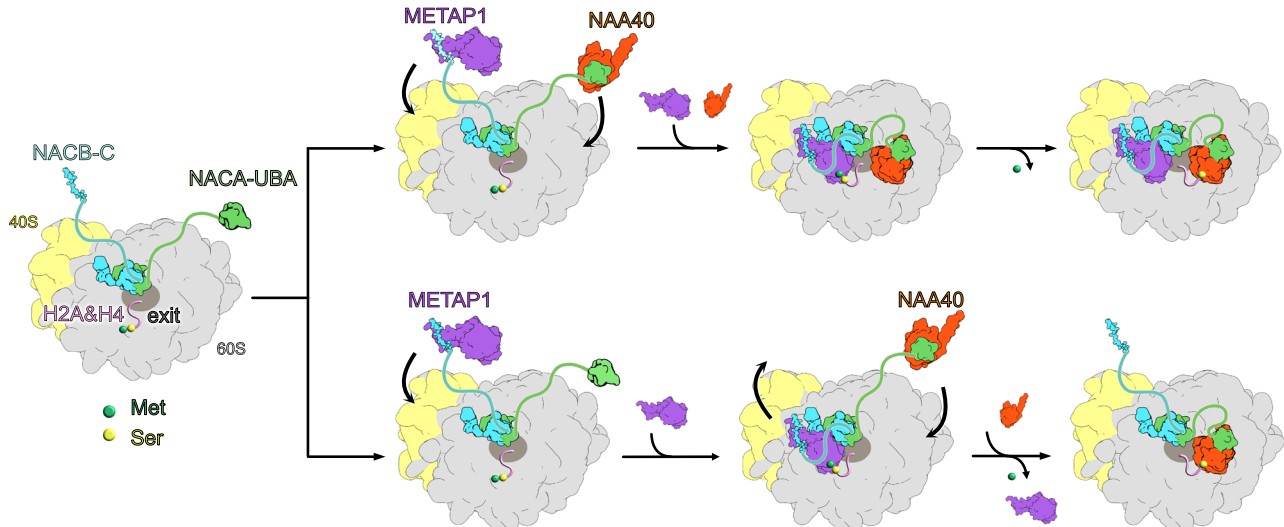

**Fig. 5 | Steps of co-translational histone Nt-acetylation by NAA40.** The top branch illustrates a concerted binding of METAP1 and NAA40 that is coordinated by NAC, which couples methionine excision and histone Nt-acetylation. The bottom branch shows an alternative stepwise ribosome association of METAP1 and NAA40 coordinated by NAC that is physically decoupled.

recruited to the ribosome. Hence, in contrast to a conservative step-by-step factor exchange mechanism our structures provide additional support for an intricate cross-talk between proteins and a coordination of modification events that appear very plausible for the highly competitive environment at the peptide exit site (Fig. 5). However, while our data further expands the inventory of co-translational modification factors forming multi-protein assemblies on the ribosome, data on these types of complexes remain mainly based on in vitro experiments[6,7]. Therefore, future investigations are needed to further interrogate the order of binding events and the formation of these transient multifactor assemblies in vivo. While in cell insights on these likely transient assemblies are crucial to understand and confirm the interplay between these different factors, investigating them by in vivo studies remains challenging. To capture these multi-protein interactions well optimized live-cell single-molecule fluorescence resonance energy transfer (smFRET) experiments or exhausting in situ cryo-EM studies are potentially required.

Our structural and biochemical data also demonstrate that the unique N-terminal helical extension of NAA40 acts as a substitute for additional auxiliary subunits, which are required for all other co-translational NATs for ribosome binding[2]. Interestingly, the positively charged α0 helix likely also functions as a nuclear localization sequence for NAA40[28]. The resulting distribution of NatD to both cytoplasm and nucleus makes it further stand out compared to most other NATs. It also poses the question of how the balance between the cytoplasmic co- and potential nuclear post-translational activity of NAA40 is regulated, since Nt-acetylation is generally considered irreversible. This is particularly important since co-translational histone H4/H2A Nt-acetylation can be considered as the earliest epigenetic mark placed. Notably, histone N-terminal acetylation has been shown to be an abundant modification, occurring on more than 90% of H2A and H4 proteins[35,36], suggesting that the elucidated co-translational N-terminal acetylation involving NAA40 and the NAC complex is a highly prevalent mechanism within cells. Beyond canonical histones, it has been recently shown that NAA40 also targets histone variant H2A.X that shares the same N-terminal residues as H4 and H2A, and plays a crucial role in DNA damage response[8]. Taking into account our native NAA40 pullout experiment in which DNA damage repair factors are co-precipitated, one possibility is that NatD might be involved in this pathway, potentially via targeting H2A.X.

Epigenetic perturbations have by now been established as drivers of oncogenesis[37]. In line with that, different studies have shown the oncogenic role of NAA40, demonstrating that its dysregulation promotes the development or progression of various types of cancer, including those of lung and colorectal, via alteration of histone H2A and H4 Nt-acetylation[13]. Accordingly, NAA40 has been suggested as an anti-cancer drug target. As our structural data provides further insights into the molecular details of co-translational NAA40 activity, it might further drug development efforts.

## Methods

### Factor-free 80S preparation
Empty 80S were prepared as described before[38]. In short, FreeStyle 293-F cells were harvested by centrifugation at $250 \, x \, g$ for 5 min and washed once with 1x PBS. The cell pellet was resuspended in HS Buffer (20 mM HEPES/KOH pH 7.5, 500 mM KOAc, 2 mM MgCl$_2$, 1 mM DTT) supplemented with 0.5% IGEPAL CA-630 (Sigma Aldrich) and incubated for 10 min. The resulting lysate was cleared by centrifugation at $2960 \, x \, g$ for 15 min and $36{,}500 \, x \, g$ for 25 min. Ribosomes were pelleted through a 1 M sucrose cushion in HS Buffer at $106{,}700 \, x \, g$ for 18.5 h using a Type Ti 70 rotor (Beckman Coulter). The ribosome pellet was dissolved in HS Buffer, and potential aggregates were removed by centrifugation at $10{,}000 \, x \, g$ for 5 min. Ribosome subunits and nascent chains were dissociated by the addition of 1 mM puromycin and subsequent incubation for 15 min on ice and 10 min at 37 °C. Subunits were separated through a linear 10–40% (w/v) sucrose gradient in HS Buffer at $202{,}496 \, x \, g$ for 2.5 h using a SW 40 Ti rotor (Beckman Coulter). Fractions corresponding to 40S and 60S were separately collected and pelleted through a 1 M sucrose cushion in LS Buffer (20 mM HEPES/KOH pH 7.5, 100 mM KOAc, 10 mM MgCl$_2$, 1 mM DTT) at $140{,}900 \, x \, g$ for 14 h in a Type Ti 70 rotor (Beckman Coulter). Subunit pellets were resuspended in LS Buffer, and factor-free 80S were formed by incubating equimolar amounts of small and large subunits for 1 h on ice. To remove remaining subunits from the newly formed 80S, the mixture was separated through a linear 10–30% (w/v) sucrose gradient in LS Buffer at $202{,}496 \, x \, g$ for 2.5 h using a SW 40 Ti rotor (Beckman Coulter). The 80S fraction was pelleted through a 1 M sucrose cushion in LS Buffer at $494{,}648 \, x \, g$ for 1 h using a TLA-110 rotor (Beckman Coulter). Finally, the pellet was resuspended in LS Buffer, frozen in liquid nitrogen and stored at -80 °C.

## NAA40-FLAG, NAA40S-FLAG and METAP1 protein purification

For NAA40 and NAA40S, Rosetta 2(DE3) (Novagen) cells transformed with plasmids encoding the proteins with an N-terminal His$_6$-SUMO and a C-terminal FLAG tag (see below). For METAP1, a plasmid encoding for His$_6$-SUMO-METAP1 (see below) was transformed in Rosetta 2(DE3) cells. For all proteins, cells were grown in LB medium to a density of 0.6–0.8 OD$_{600}$ at 37 °C. Then, protein expression was induced with 0.1 mM IPTG at 18 °C overnight. Cells were harvested by centrifugation, washed once with 1x PBS and resuspended in Lysis Buffer (50 mM HEPES pH 7.5, 500 mM NaCl, 5% glycerol, 10 mM imidazole, 1 mM DTT, 10 μg/mL DNaseI) before lysis in a continuous flow cell disruptor (Constant Systems). Cell lysates were cleared by centrifugation at $36,500 \, xg$ for 30 min and the cleared lysates were incubated with cOmplete Ni-metal affinity beads (Roche) for 1 h. Then beads were washed 3 times with HS-buffer (50 mM HEPES/NaOH pH 7.5, 500 mM NaCl, 10 mM imidazole, 5% glycerol, imidazole, 1 mM DTT) and 3 times with LS-buffer (50 mM HEPES/NaOH pH 7.5, 100 mM NaCl, 10 mM imidazole, 1 mM DTT). NAA40-FLAG and METAP1 were eluted in Elution Buffer (20 mM HEPES/NaOH, pH 7.5, 100 mM NaCl, 5% glycerol, 0.1 mg/mL SUMO-protease (homemade), 10 mM imidazole, 1 mM DTT) overnight.

NAA40S-FLAG was eluted with I-Elution Buffer (20 mM HEPES/NaOH, pH 7.5, 100 mM NaCl, 5% glycerol, 300 mM imidazole, 1 mM DTT). The tag was removed by dialysis in the presence of 0.01 mg/mL SUMO-protease (homemade) against Dialysis Buffer (20 mM HEPES/NaOH, pH 7.5, 100 mM NaCl, 10 mM imidazole, 1 mM DTT) overnight. SUMO protease was removed by running the sample over cOmplete Ni-metal affinity beads (Roche).

For both NAA40-FLAG and METAP1, elution fractions were applied to a HiTrap Q FF column (GE Healthcare). Proteins were eluted with a linear gradient from 100-500 mM NaCl in Ion Exchange Buffer (20 mM HEPES/NaOH, pH 8.0, 100 mM NaCl, 1 mM DTT). Protein-containing fractions were pooled, and buffer was exchanged using PD-10 desalting columns (Cytiva) to Storage Buffer (20 mM HEPES/KOH, pH 7.5, 100 mM KOAc, 5 mM MgCl$_2$, 1 mM DTT.

NAA40S-FLAG was further purified exactly like NAA40-FLAG, but instead of an anion exchange column, a HiTrap SP FF column (GE Healthcare) was used.

## NAC-wt and -ΔUBA purification

The genes coding for the human NAC heterodimer consisting of N-terminally His$_6$-thrombin cleavage site-tagged NACA and NACB isoform 2 were expressed from pET28a (Novagen) (see below). Freshly transformed *Escherichia coli* ER2566 (NEB) were grown in LB medium at 37 °C to an OD$_{600}$ of 0.8, induced by the addition of 1 mM IPTG and continued to grow for 3 h. Cells were lysed using a cell disruptor (Constant Systems Ltd.) at 30 Kpsi in lysis buffer (20 mM HEPES, pH 8.0, 500 mM NaCl, 0.1 mM PMSF, 20 μg/ml DNaseI and 1× cOmplete EDTA-free protease inhibitor cocktail (Roche). Lysates were clarified by centrifugation in a A27-8 × 50 rotor (Thermo Scientific) in a LYNX6000 centrifuge (Sorvall) at $36,500 \, xg$ for 25 min. The supernatant was incubated with Protino Ni-NTA Agarose slurry (Macherey & Nagel) equilibrated in lysis buffer for 30 min at 4 °C. Beads were transferred to a gravity flow column, washed with 40 column volumes of lysis buffer and eluted with 5 column volumes of lysis buffer containing 500 mM imidazole. NAC-containing fractions were pooled and dialyzed (MWCO 12–14 kDa) overnight against buffer B (20 mM HEPES, pH 8.0, 50 mM NaCl, 2 mM CaCl$_2$) in the presence of 20 units of thrombin (Merck) to remove the His$_6$-tag. Dialyzed samples were loaded on a 5 ml HiTrap SP HP column (Cytiva) pre-equilibrated with 20 mM HEPES pH 7.0, 50 mM NaCl for removal of excess NACA and eluted with a 6 column volume linear gradient to 100% Buffer C (20 mM HEPES pH 7.0, 1 M NaCl). Heterodimeric NAC eluted at around 450 mM NaCl. PD10 columns (Cytiva) were used to transfer heterodimeric NAC fractions into the final Buffer D (20 mM HEPES pH 7.5, 100 mM KOAc, 5 mM MgCl$_2$, 5% glycerol and 1 mM DTT). Protein concentration was determined using the molar extinction coefficient of human NAC (ε280 = 2980 M$^{-1}$ cm$^{-1}$).

The NAC-ΔUBA mutant was expressed and purified in the same way, with the exception that after the HiTrap SP HP column, NAC-containing fractions were pooled, concentrated using Amicon Ultra Centrifugal Filter (10 kDa Mw cutoff, Merck) and loaded on a HiLoad Superdex 75 pg 16 mm column (Cytiva) using Buffer E (20 mM HEPES/KOH, pH7,5, 100 mM KOAc, 5 mM MgCl$_2$, 1 mM DTT). NAC-containing fractions were pooled.

## In vitro binding assay

Six picomoles 80S were incubated with 60 pmol of NAC, NAC-ΔUBA, NAA40-FLAG, NAA40S-FLAG or METAP1 in Binding Buffer (20 mM HEPES/KOH, pH7,5, 100 mM KOAc, 5 mM MgCl$_2$, 1 mM DTT) for 1.5 h. Mixtures were incubated with anti-FLAG M2 affinity gel (Sigma Aldrich) for 1 h. Beads were washed three times with Binding Buffer and eluted with Binding Buffer supplemented with 0.25 mg/mL 3xFLAG peptide (Sigma Aldrich) for 1.5 h. Input, flow-through and elution fractions were TCA precipitated and resuspended in Laemmli Buffer. Fractions were analyzed by SDS-PAGE.

## Cell culture

Adherent HEK293T cells were cultured in DMEM (Gibco) supplemented with 1x GlutaMAX (Gibco), 1x Penicillin-Streptomycin (Gibco) and 10% FBS (Gibco) at 37 °C and 5% CO$_2$. FreeStyle 293-F suspension cells were grown in HyClone HyCell TransFx-H (Cytiva) supplemented with 3× GlutaMAX (Gibco), 1x Penicillin-Streptomycin (Gibco) and 0.01% Poloxamer 188 (Sigma Aldrich) in a Multitron Cell shaker (Infors HT) at 37 °C, 80% humidity and 5% CO$_2$. Adherent HCT116 cells were cultured in McCoy's 5a medium (Gibco) supplemented with 10% FBS (Gibco, Invitrogen) and 1% Penicillin-Streptomycin (Gibco). Cells were grown in a humidified atmosphere at 37 °C containing 5% CO$_2$. The HCT116 cell line was used to generate the NAA40−/− cell line (NAA40-KO) using CRISPR, as previously described[29], or for the ectopic expression of wild-type NAA40-V5, as previously described[39].

## Native NAA40 ribosome complex isolation

Half confluent HEK293T cells were transiently transfected with 0.5 μg/mL pcDNA5/FRT/TO-NAA40-3C-3xFLAG plasmid for 24 h using PEI. Then, cells were harvested by scraping, pelleted at $250 \, xg$ for 5 min and washed once with 1x PBS. The cell pellet was resuspended in Lysis Buffer (20 mM HEPES pH 7.5, 150 mM KOAc, 5 mM MgCl$_2$, 1 mM DTT, 0.1% IGEPAL CA-630 (Sigma Aldrich), 1× cOmplete EDTA-free Protease Inhibitor Cocktail (Roche)) and sonicated with four 10 s pulses and 30 s on ice in between. The lysate was cleared by two consecutive centrifugation steps at $2960 \, xg$ for 15 min and $36,500 \, xg$ for 25 min, respectively, followed by incubation for 1.5 h with anti-FLAG M2 affinity gel (Sigma Aldrich) pre-equilibrated in Lysis Buffer. After beads were washed twice with Wash Buffer (20 mM HEPES pH 7.5, 150 mM KOAc, 5 mM MgCl$_2$, 1 mM DTT, 0.01% IGEPAL CA-630) and once with Elution Buffer (20 mM HEPES pH 7.5, 150 mM KOAc, 5 mM MgCl$_2$, 1 mM DTT, 0.05% octaethylene glycol monododecyl ether) in batch, they were transferred to a 1 mL Mobicol (MoBiTec) spin column. Then, the beads were washed once more with Elution Buffer before elution for 1.5 h at 4 °C in Elution Buffer supplemented with 0.25 mg/mL HRV 3C protease (homemade). The elution was collected by centrifugation at $450 \, xg$ for 1 min and either directly used for cryo-EM grid preparation or subjected to SDS-PAGE. Prominent protein bands were excised and subjected to mass spectrometry (MS) analysis by our in-house MS facility (Supplementary Data 1) to identify and annotate the respective proteins in Fig. 1a.

### NAA40-NAC-METAP1-80S in vitro reconstitution

One hundred eighty nanomolar factor-free 80S were incubated with 1.08 μM NAC in Reconstitution Buffer (20 mM HEPES pH 7.5, 100 mM KOAc, 5 mM MgCl$_2$, 1 mM DTT, 0.05% octaethylene glycol mono-dodecyl ether) for 10 min, followed by the addition of 1.08 μM METAP1 and incubation for 10 min. Lastly, 1.08 μM NAA40-FLAG were added and incubated for another 30 min. Then, the reconstitution mixture was directly subjected to cryo-EM grid preparation.

### Cryo-EM sample preparation

For all samples, 3.5 μL were applied to R3/3 copper grids with an additional 3 nm carbon layer (Quantifoil) using a Vitrobot (FEI) at 5 °C and 85–90% humidity. Grids were blotted for 3 s with 0 blot force and 45 s pre-blot time before plunge freezing in liquid ethane.

### Cryo-EM data collection and processing

All cryo-EM data sets were collected on a Titan Krios (FEI) at 300 kV equipped with a Falcon 4i direct electron detector (Thermo Fisher) and a Selectris X energy filter (Thermo Fisher) at 5 eV slit width using EPU (v3.3.1) (Thermo Fisher). A nominal pixel size of 0.727 Å/pixel, a total dose of 40 e⁻/Å$^2$ and a defocus range of 0.5–3.5 μm were used. Motion correction was performed using MotionCor2[40], and initial CTF parameters were determined using CTFFIND4[41]. Micrographs with a resolution worse than 5 Å were removed.

For the native NAA40-NAC 80S sample, a total of 42,123 micrographs were used, and particles were picked using crYOLO[42] (v1.7.6) using the general model on lowpass filtered micrographs (Supplementary Fig. 1). Then, particles were extracted with 160 pixels box size, four times binned in Relion[43] (4.0.1) and imported in cryoSPARC[44] (v4.4.0) for 2D classification. Classes corresponding to 80S ribosomes (306,751 particles) were selected and refined using homogenous refinement with an ab initio generated volume as a reference. After 3D classification, all ribosome classes with distinct exit density were combined and refined together (239,101 particles). Using a small spherical mask, particles were classified for the presence of NAA40 and in a second classification step, sorted for the presence of stable NAA40-NAC complexes using a wider mask around the peptide exit in cryoSPARC. The resulting particles were re-imported into Relion and further classified with a mask around NAC. Particles with good density for NAC and NAA40 (47,197 particles) were re-extracted with an unbinned box size of 640 pixels. The un-binned particles resembling an inactive translation state were refined in cryoSPARC using homogenous refinement with global CTF and per particle defocus refinement to a resolution of 2.67 Å in cryoSPARC and subsequently local resolution filtered (Supplementary Fig. 2a–d upper panels, Supplementary Table 1). To isolate active ribosomes, a new round of unmasked 3D classification was performed in cryoSPARC, starting from the initial 3D refinement in cryoSPARC. A class with both tRNAs in the intersubunit space and density for NAC-NAA40 at the exit was selected (66,717 particles). Particles were then sorted for the presence of NAC-NAA40 with a wide mask around the peptide exit site. Particles containing stable NAC-NAA40 density (28,948 particles) were further sorted for their translational state with a mask around the intersubunit space, resulting in a defined P/P E/E POST-state (19,178 particles). The final active 80S-NAA40-NAC particles were re-extract un-binned with a box size of 640 pixels and refined using homogenous refinement with global CTF and per particle defocus refinement in cryoSPARC to a resolution of 2.91 Å (Supplementary Fig. 2a–d, lower panels and Supplementary Table 1), followed by local resolution filtering. To obtain nucleosome particles, classes excluded after the initial 2D classification were subjected to another round of reference-free 2D classification, which resulted in a few 2D classes representative for nucleosome side views (Supplementary Fig. 1; 11,799 particles). However, further processing or particle enrichment efforts were unsuccessful.

For the in vitro reconstituted METAP1-NAA40-NAC-80S sample, a total of 63,237 micrographs were collected. Initial particles were picked using crYOLO[42] (v1.7.6) with the general model on low-pass filtered micrographs (Supplementary Fig. 7). Particles were extracted four times, binned with a box size of 160 pixels in Relion[43] (v4.0.1) and imported into cryoSPARC[44] (v4.4.0) for 2D classification. Good 80S classes were selected (805,997 particles) and 3D refined using an ab initio reference. After 3D classification, particles corresponding to 80S with additional density at the peptide exit were selected (466,068 particles) and further sorted for exit site density with a wide mask. The resulting particles (167,604 particles) were re-imported to Relion and sorted for the presence of METAP1 with a spherical mask around NAA40 and the METAP1 binding site. Particles lacking METAP1 (72,590) were further classified with a wide spherical mask around the peptide exit site to obtain good density for NAA40-NAC (26,095 particles). The NAA40-NAC-80S particles were re-extract in Relion with an unbinned box size of 640 pixels. They were re-imported into cryoSPARC and refined using homogenous refinement with global CTF and per particle defocus refinement to a resolution of 2.72 Å (Supplementary Fig. 8a–d lower panels and Supplementary Table 1), followed by local resolution filtering. A class containing density for METAP1 from the second focused 3D classification (Supplementary Fig. 7) was further classified in Relion with a spherical mask around METAP1, resulting in two different conformations of METAP1. Each class was then separately classified again with a wide spherical mask around the exit for stable density, resulting in 11,809 particles for METAP1-State1 (S1) and 15,772 particles for METAP1-State2. For both states, particles were re-extracted with an unbinned box size of 640 pixels and refined in cryoSPARC using homogeneous refinement with global CTF and per particle defocus refinement to resolutions of 3.54 Å for METAP1-State1 (Supplementary Fig. 8a–d upper panels and Supplementary Table 1) and 3.44 Å for METAP1-State2 (Supplementary Fig. 8a–d middle panels and Supplementary Table 1). Finally, volumes were local resolution filtered.

All cryo-EM densities were visualized using ChimeraX[45] (v1.9).

### Model building

For the molecular model of the ex vivo combined inactive 80S-NAA40-NAC reconstruction, PDB:6Z6M[20] was used as the base model for the ribosome, SERBP1 and eEF2. The ribosome-bound NAC globular domain and NACB ribosome anchor were taken from PDB:7QWR[27] and rigidly docked into the densities. For NAA40 with the NACA-UBA domain, our AF2 multimer model was used and a rigid body fit.

For the model of the ex vivo active 80S-NAA40-NAC reconstruction, the ribosomal model from PDB:6Y2L[46] was used. Models for ribosomal proteins eS12 and eS31 were taken from PDB:8GLP[47]. The P- and E-site tRNAs from PDB:6GZ5[19] were used as placeholders for our tRNAs. Placeholders for mRNA and nascent chain were built de novo. For the NACB ribosome anchor and the NAC globular domain PDB:7QWR[27] was used again, and both NAC parts were independently fit initially. NAA40 with the NACA-UBA domain from our AF2 multimer model were rigidly docked.

For all in vitro reconstituted 80S complexes, the initial factor-free 80S model from PDB:8YOO[38], the AF2 multimer prediction for NAA40 with the NACA-UBA domain and the AF2 multimer prediction for the NAC globular domain and the NACB ribosome anchor were used. All ribosome-bound factors were rigidly docked. For 80S-NAA40-METAP1-NAC State1 and State2, the AF2 database[23] model for METAP1 was rigid-body fitted, and the zinc finger domain was removed manually.

All models were real-space refined in Phenix[48] (v1.19.2-4158 and v1.21.2-5419) and manually adjusted in Coot[49] (v0.9.8). Molecular models were visualized with ChimeraX[45] (v1.9).

### RNA extraction and quantitative Real Time PCR (qRT-PCR)

Total RNA was extracted using the RNeasy Mini kit (Qiagen) according to the manufacturer's instructions. An amount of 0.5 µg total RNA was then reverse transcribed to complementary DNA using the PrimeScript RT reagent kit (Takara) with random primers. qRT-PCR was carried out using KAPA SYBR Green (SYBR Green Fast qPCR Master Mix) and the Biorad CFX96 Real-Time System. Expression data were normalized to the mRNA levels of the β-actin housekeeping gene and calculated using the $2^{-\Delta\Delta Ct}$ method. Primer sequences were obtained from IDT (Supplementary Table 2).

### Protein extraction

For whole-cell protein extraction, cell pellets were resuspended in 1× PBS at a ratio of two volumes of 1x PBS to one volume of cell pellet. Protein extracts were obtained by adding by adding an equal volume of 2× lysis buffer to the resuspended cells (0.1 M Tris-HCl, pH 7, 4% SDS, and 12% β-mercaptoethanol), resulting in a final 1x concentration. The samples were then boiled at 95 °C for 10 min, followed by centrifugation at 12,000 $x g$ for 10 min at 4 °C to isolate proteins. Total protein concentration was quantified using a Nanodrop ND-1000. For histone acid extraction, cells were lysed in hypotonic lysis buffer (10 mM Tris-HCL, pH 8, 1 mM KCL, 1.5 mM MgCl₂, 0.1% Triton X-100 and 1x protease inhibitor cocktail) and incubated for 30 min with constant agitation at 4 °C. Isolated nuclei were then washed once in hypotonic lysis buffer and, after centrifugation at 6500 $x g$ for 10 min, were resuspended in 0.2 M HCL ($4 \times 10^7$ nuclei per mL) and incubated overnight with constant rotation at 4 °C. Histones were isolated by centrifugation at 6500 $x g$ for 10 min, and the pH was neutralized with 2 M NaOH at 1/10 of the volume of the supernatant.

### Biochemical fractionation

Ten million cells were harvested in 1x PBS and lysed in Buffer S (10 mM HEPES, 10 mM KCL, 1.5 mM MgCl₂, 0.34 mM sucrose, 10% glycerol) plus 0.1% Triton X-100 and 1x protease inhibitor cocktail) on ice for 10 min. Following centrifugation at 1300 $x g$ for 5 min at 4 °C, the supernatant S1 was centrifuged at maximum speed for 10 min, and the supernatant S2 was taken as the cytoplasmic fraction. Pellet was washed with 5× pellet volume nuclear extraction Buffer A (10 mM HEPES, 1.5 mM MgCl₂, 10 mM KCl) plus 100 mM NaCl and 1 mM DTT, followed by centrifuge at 3000 $x g$ for 5 min at 4 °C.

In order to purify the nuclear fractions, pellets were then resuspended in a low-salt lysis buffer (20 mM HEPES, 1.5 mM MgCl₂, 150 mM NaCl, 0.2 mM EDTA, 25% glycerol) plus 0.1% v/v NP-40, 0.2 mM DTT and protease inhibitors (0.5 mM PMSF and 1x protease inhibitor cocktail) and treated with 200 units/mL of benzonase (Sigma E1014) for 15 min at 37 °C. Following sonication for $2 \times 5$ s pulses with a Bioruptor (Diagenode), samples were centrifuged at 16,000 $x g$ for 10 min at 4 °C to obtain the supernatant that represents the low- salt nuclear fraction.

### Immunoblotting

Twenty micrograms of protein extract or three micrograms of histone extract were separated on SDS-PAGE and then transferred to a nitrocellulose membrane (GE Healthcare). After blocking with 3% TBS-T/BSA for 1 h at RT, the membranes were incubated with the primary antibodies overnight at 4 °C. The primary antibodies used were as follows: H2A/H4S1Ph (1:1,000, Abcam, ab177309, LOT: GR214941-1, Clone EPR18184), H4 (1:1,000, Millipore, 05-858, LOT: 3836545, Clone 62-141-13), H2A (1:1,000, Millipore, 07-146, LOT: 1970740, Polyclonal), ACTIN (1:1,000, Abcam, ab8227, LOT: 1065644-1, Polyclonal), V5 (1:1,000, Abcam, ab27671, LOT: GR322548-8, Clone SV5-Pk1), GAPDH (1:2,000, Abcam, ab9485, LOT: GR3318490-1, Polyclonal), NAA40 (1:1,000, Abcam, ab106408, LOT: GR3409009-8, Polyclonal), FLAG (1:1,000, Sigma-Aldrich, F1804, Clone: M2). For secondary antibody, a horseradish peroxidase (HRP)-conjugated goat anti-rabbit IgG (1:30,000, Thermo Scientific, 31462, LOT: 095113I, Polyclonal), or an

HRP-conjugated goat anti-mouse IgG (1:20,000, Thermo Scientific, A16078, LOT: 86-132-110221, Polyclonal) was used.

### Transient RNA interference

HCT116 cell lines were seeded in antibiotic-free medium and grown to 60% confluence at the time of transfection. Subsequently, the cells were transiently transfected with 50 nM of siNACA (L-027161-00-0005, Horizon discovery) or the negative control (D-001810-10-05, Horizon discovery) for 72 h using DharmaFECT 1 Transfection Reagent (Horizon discovery) according to the manufacturer's instructions.

### Transient exogenous expression for NACA complementation

Human siRNA-resistant pcDNA5/FRT/TO-FLAG-NACA and pcDNA5/FRT/TO-FLAG-NACA-ΔUBA plasmids were transfected into HCT116 cells. In brief, cells were seeded in antibiotic-free medium and grown to 60% confluence at the time of the transfection. Cells were transfected with 2 µg plasmid DNA carrying human FLAG-NACA or FLAG-NACA ΔUBA using FuGENE HD (Promega Corporation) according to the manufacturer's instructions. Cells were harvested 48 h post-transfection.

### Generation of Nt-Ac antibody

Rabbit polyclonal antibodies were raised in collaboration with LifeTein LLC (USA). To generate the Nt-Ac antibody, the animal was immunized with a Keyhole Limpet Hemocyanin-conjugated synthetic tetravalent Multiple Antigenic Peptide (MAP4-C) containing the acetylated sequence Ac-SGRGK on a lysine-based core. The peptides used were not biotinylated. Serum was affinity-purified by the manufacturer against peptides with and without N-terminal acetylation to enrich for recognizing N-terminal acetylation. After the purification, antibody specificity was determined by enzyme-linked immunosorbent assay using the specific and non-specific peptides as antigens.

### Dot blot analysis

Synthesized biotinylated peptides with at least 95% purity (BIO-SYNTAN GmbH) were dissolved in water, and drops containing 25, 20, or 15 µM were deposited on a PVDF membrane, and allowed to air-dry for 1 h. The membrane was then submerged in 100% methanol for 1 min and 1× PBS for another minute. After blocking with 3% TBS-T/BSA for 1 h at RT, the membranes were incubated with the primary antibodies overnight at 4 °C. The primary antibodies used were as follows: Biotin (1:1,000, Santa Cruz, sc-101339, LOT: J1019, Clone 33) and the Nt-Ac antibody (1:1,000) generated in collaboration with LifeTain LLC (USA). For secondary antibody, an HRP-conjugated goat anti-rabbit IgG (1:30,000, Thermo Scientific, 31462, LOT: 095113I, Polyclonal) was used.

### Molecular cloning

For overexpression of NAA40-3C cleavage site-3 × FLAG in human cells, the coding sequence was amplified from cDNA and cloned into a pcDNA5/FRT/TO-3C-3 × FLAG plasmid using BamHI and XhoI restriction sites. For expression of FLAG-NACA and -NACA-ΔUBA expression in human cells, the NACA coding sequence or the ΔUBA fragment was amplified from cDNA and cloned into a pcDNA5/FRT/TO-FLAG plasmid using BamHI and XhoI restriction sites. For siRNA-resistant NACA and NACA-ΔUBA expression constructs, fragments from the respective pcDNA5 plasmids were amplified with mutations and complementary overhangs by PCR and assembled using Fast Cloning[50]. For recombinant purification of NAA40- and NAA40S-FLAG, the NAA40 coding sequence and a C-terminal FLAG-tag were simultaneously inserted into a pET-28a plasmid (Novagen) with an N-terminal His₆-SUMO tag using Fast Cloning[50]. NAA40S was inserted the same way into the expression plasmid. For recombinant purification of METAP1, the coding sequence was inserted into a pET-28a plasmid (Novagen) with an N-terminal His6-SUMO tag using FastCloning[50]. For recombinant

purification of NAC or NAC-ΔUBA complexes, the sequences for NACA or NACA-ΔUBA (1–173aa) were amplified and cloned into a pET-28a plasmid (Novagen) with an N-terminal His$_6$-thrombin cleavage site-tag using NdeI and BamHI restriction sites. In the resulting plasmids, the coding sequence for NACB isoform 2 with a preceding ribosome binding site was cloned into the same vector using EcoRI and NotI restriction sites. Primer sequences are listed in Supplementary Table 2.

## Statistics and reproducibility
Representative biochemical experiments were repeated at least once with highly similar results. Cryo-EM data for ex vivo and in vitro samples were collected once. For quantification of Western Blots and knockdown validation by qPCR, statistical tests and number of biological replicates are indicated. Mean values and individual data points are displayed. Error bars representing the standard error of the mean (SEM) are shown.

## Reporting summary
Further information on research design is available in the Nature Portfolio Reporting Summary linked to this article.

## Data availability
All cryo-EM maps and molecular models were deposited to the Electron Microscopy Data Bank (EMDB) and the Protein Data Bank (PDB), respectively. They are accessible under the following codes: PDB: 9T6D and EMD-55613 (NAA40-NAC bound human 80S (combined translation states)); PDB: 9T6C and EMD-55612 (NAA40-NAC bound active human 80S); PDB: 9RG7 and EMD-53944 (in vitro reconstituted NAA40-NAC bound 80S); PDB: 9T69 and EMD-55610 (in vitro reconstituted METAP1 -NAA40-NAC bound 80S State 1); PDB: 9T6I and EMD-55616 (in vitro reconstituted METAP1-NAA40-NAC bound 80S State 2). Source data are provided with this paper.

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

## Acknowledgements

We thank C. Ungewickell and S. Rieder for technical assistance. We thank Thomas Fröhlich and the Laboratory for Functional Genome Analysis (LAFUGA) for MS analysis and interpretation. This study was supported by grants from the ERC (ADG 885711 Human-Ribogenesis and DFG (BE1814/20-1, BE1814/22-1) to R.B., D.G., and T.D. were supported by the Graduate School of Quantitative Biosciences Munich (QBM). D.G. was also supported by the LMU—China Scholarship Council (CSC). Research work in the A.Ki. lab was supported by the European Regional Development Fund and the Republic of Cyprus through the Research & Innovation Foundation (Project: EXCELLENCE/0421/0152) and a Cyprus Cancer Research Institute's (C.C.R.I) Bridges in research excellence grant (CCRI_2020_FUN_001-103) under agreement No. CCRI_2021_FA_LE_106.

## Author contributions

D.G, T.D., A.Kl, B.B., A.Ki, and R.B. conceived the study. A.Kl. performed in vivo experiments. D.G. performed in vitro experiments. D.G. prepared cryo-EM samples with the help of T.D. Cryo-EM data were collected by O.B. and processed by D.G. Molecular models were prepared by D.G. and M.T. Structural data were interpreted by D.G., T.D., and R.B., and visualized by D.G. The manuscript was written D.G., T.D., A.Kl, A.Ki, B.B., and R.B. All authors revised and commented on the manuscript.

## Funding

## Competing interests

The authors declare no competing interests.
