## [Transparent Peer Review file · Nature Communications]

NAA40 and NAC cooperate in co-translational histone acetylation in humans

Corresponding Author: Professor Roland Beckmann

Version 0:

Reviewer comments:

Reviewer #1

(Remarks to the Author)

Remarks to the authors:

Guan et al. combine immunopurification, cryo-electron microscopy, and biochemical assays to reveal how NAA40 engages the ribosome during translation. They show that NAA40 associates with ribosomes in vivo and that this interaction depends on the nascent polypeptide-associated complex (NAC). Structural analyses place NAA40 and NAC adjacent to the peptide exit tunnel, where the positively charged $\alpha 0$ helix of NAA40 functions as a ribosome anchor in a manner reminiscent of NatA/E. Functional experiments demonstrate that the UBA domain of NACA is essential for stable NAA40 recruitment and efficient histone acetylation. Moreover, reconstitution assays reveal that NAA40 can form a multienzyme assembly with methionine aminopeptidase 1 (MetAP1) and NAC, suggesting coordinated removal of the initiator methionine and subsequent acetylation. Together, the authors delineate the molecular architecture and cofactors underlying co-translational histone N-terminal acetylation by NAA40 and highlight the dual potential of NAA40 for both co- and post-translational activity.

This study provides new mechanistic insights into NatD function and establishes NAC as a general recruiter of co-translational modifying enzymes. The work will be of interest to structural biologists, by extending our knowledge of ribosome-bound modification factors; to epigeneticists, by clarifying the timing and regulation of histone acetylation; and to the broader field of protein biogenesis and translational control, by illustrating how multiple enzymatic activities are coordinated at the ribosomal exit site. The manuscript is clearly written and the figures well presented. We recommend publication after the following points are addressed.

General points:

While the structural and biochemical data convincingly reveal how NAA40 can be recruited to the ribosome and coordinated with NAC and METAP1, much of the evidence relies on in vitro reconstitution. The authors do point this out in the text (e.g. lines 254–257, 280–283), but it would be helpful if they more clearly emphasized the current lack of full in vivo confirmation and discussed how future work might address this.

Minor points

1. Please check whether referencing in the abstract complies with Nat. Comms. guidelines.
2. Line 49: write out the abbreviations “Ph” and “me” in full (or in parentheses) at first mention.
3. Fig. 1a: Clarify how histones, XRCC5/6, and PRKDC were identified (e.g. MS, immunoblotting) and provide supporting data. In addition, there is no negative control IP (e.g. FLAG-empty, untagged, or unrelated FLAG-protein) to assess specificity.
4. Line 88-90 and line 92 is referenced to Supplementary Fig. 1a and b, but should be 2a and b. In the text is written AF2, in the figure legend AF3.
5. Fig. 3a: there is a quite striking difference between lane 7 and 9. From the presented data, it seems likely, NAC Δ UBA occludes NAA40's weak direct binding to the ribosome. This could be tested by adding increasing amounts of NAC-deltaUBA to see if it displaces NAA40 from the ribosomes. Alternatively, titrating NAC vs. NAC-deltaUBA would strengthen the case that UBA-dependent binding is specific, not an artefact. Rephrase then the statement that NAC- Δ UBA “abolishes” ribosome binding accordingly. The assay is qualitative; additional controls would strengthen the data (e.g. NatA) or quantification or maybe mutants in the N-terminus of NAA40. Shift the legend in 3a slightly upward for clarity.
6. Antibody validation (Fig. 5e): Were the peptides biotinylated? It is nowhere stated in the methods and not clear. Was the antibody further purified to ensure Ac-binding, e.g. against free-Nt peptides binding and the FT against Ac-Nt binding?
7. Line 186: Residual Nt-acetylation in NACA-KD cells may also result from incomplete knockdown.
8. There is missing one H.S. in Supplementary Fig. 5h

Reviewer #2

(Remarks to the Author)

Reviewer #3

(Remarks to the Author)

This study provides compelling structural evidence for NAC-mediated recruitment of NAA40 to ribosomes. Combined with biochemical data demonstrating that NACA knockdown reduces H4/H2A acetylation, these findings significantly advance our understanding of co-translational histone modification. Overall, this is an excellent study with high quality in experimental data and data presentation. However, the following major and minor issues require clarification in order to strengthen the manuscript.

Major Issues:

1. NAA40 binds both hibernating and active ribosomes, yet the cryo-EM structures show no engagement of its catalytic domain with the cognate substrate motif (Ser-Gly-Arg-Gly). This may arise from non-physiological NAA40 overexpression, permitting indiscriminate ribosome association irrespective of nascent chain identity. To ensure substrate binding ability of NAA40, I recommend the authors to perform AlphaFold-Multimer predictions of NAA40 bound to H2A/H4 N-termini and check whether this binding mode conflicts with resolved NAA40-ribosome interactions.
2. The NAA40 α 0 helix mediates ribosome anchoring via rRNA (H19/H24/H46)—an interaction that is nascent chain-independent and topologically similar to NatA/E complexes. Given that NATs compete for overlapping ribosomal sites, I recommend the authors to discuss how NAC orchestrates histone-specific modification despite non-specific ribosome anchoring of NAA40.

Minor Issues:

1. Include pLDDT coloring and PAE metrics for all AlphaFold-predicted interactions (e.g., NAA40-NAC interfaces) to validate model confidence.
2. "Horseradish peroxide" should be corrected to horseradish peroxidase (HRP) in Methods.

Reviewer #4

(Remarks to the Author)

The paper provides novel structural insights into the interaction of the shortest Nat protein NatD to the ribosome. In contrast to other Nats this protein consists of a single subunit, NAA40, which specifically acetylates Histones H2A and H4 after their processing by MetAP1. NAA40 thereby is recruited close to the ribosome exit tunnel by interaction with NAC in a similar manner as shown previously for NatA and NatE. A critical role in recruitment seems to play the highly charged N-domain of NAA40 and the NACA Uba domain. Based on their structural data, the authors suggest that MetAP1 and NAA40 might be recruited concomitantly to the exit tunnel, possibly indicating a coordinated cleavage and acetylation reaction. The conclusions drawn from the structural data are supported by a variety of different biochemical experiments. Both, the structural data and the biochemical data, are of very high quality and seem to be consistent with the proposed mechanisms. The manuscript is very well written and most of the Figures are very well prepared. Together with the importance of the NAA40 for its possible function in various tumors and our still limited understanding of the N-terminal modification reactions I strongly recommend this manuscript for publication in Nature Communications. There are only some minor queries that should be addressed by the authors.

I would appreciate, if the authors could discuss the unique substrate specificity of NAA40 for H2A and H4 in some more detail.

Given the copurification of XRCC5, XRCC6 and PRKDC a more global analysis of the extract shown in Fig. 1a by qMS might provide deeper and novel insights into the substrate specificity of NAA40. It would also be interesting if gammaH2A is present in the sample.

line 289 and 290: If the N-terminal helix of NAA40 is a major determinant for ribosome binding and also harbors the nuclear localization signal, it is unlikely that the N-terminally truncated version serves in "potential nuclear post translational activity". The authors should better explain what they mean here.

Minor:

- 1: some of the Figures are too small to really see details and should be enlarged by the authors. This includes Figure 2a,3g, 4c, Extended Data Fig 1e, Extended Data Fig 2e, Extended Data Figure 3g and 3h and the Extended Data Figure 6b to 6e.
- 2: On top of Figure 3a the legends table needs to be adjusted.

3: Not sure what the authors mean with the question mark in Figure 3g. This should be removed or described in the legend.
4: Figure 4a and b. I suppose the shown maps are filtered, which should be indicated in the text of the legend.
5: Figure 5: the colors of NACB and A are too pale and should be bit darkened to provide better contrast to the gray large subunit.
6: Legend to Extended Data Figure 3: (line 797) f should be substituted by h
7: there happened some misassignment of Figures in the text: e.g. line 89/90 Extended Data Figure 1a actually refers to 2a: line 95 Extended Data Fig. 1a...;
8: I suppose the sentence in lines 214/215 refers to the experimental data in Fig. 3 and not to the structural data as indicated.
9: Line 140: references missing for NatA/E

Version 1:

Reviewer comments:

Reviewer #1

(Remarks to the Author)

The authors have satisfactorily addressed our concerns, and the manuscript is suitable for publication in Nature Communications.

Reviewer #2

(Remarks to the Author)

Reviewer #3

(Remarks to the Author)

The authors have addressed my concerns. Overall, this is a very important study and warrants the rapid publication.

Reviewer #4

(Remarks to the Author)

The manuscript significantly improved. I do not have further queries.

Point-by-point response

REVIEWER COMMENTS

Reviewer #1 (Remarks to the Author):

Remarks to the authors:

Guan et al. combine immunopurification, cryo-electron microscopy, and biochemical assays to reveal how NAA40 engages the ribosome during translation. They show that NAA40 associates with ribosomes *in vivo* and that this interaction depends on the nascent polypeptide-associated complex (NAC). Structural analyses place NAA40 and NAC adjacent to the peptide exit tunnel, where the positively charged $\alpha 0$ helix of NAA40 functions as a ribosome anchor in a manner reminiscent of NatA/E. Functional experiments demonstrate that the UBA domain of NACA is essential for stable NAA40 recruitment and efficient histone acetylation. Moreover, reconstitution assays reveal that NAA40 can form a multienzyme assembly with methionine aminopeptidase 1 (MetAP1) and NAC, suggesting coordinated removal of the initiator methionine and subsequent acetylation. Together, the authors delineate the molecular architecture and cofactors underlying co-translational histone N-terminal acetylation by NAA40 and highlight the dual potential of NAA40 for both co- and post-translational activity.

This study provides new mechanistic insights into NatD function and establishes NAC as a general recruiter of co-translational modifying enzymes. The work will be of interest to structural biologists, by extending our knowledge of ribosome-bound modification factors; to epigeneticists, by clarifying the timing and regulation of histone acetylation; and to the broader field of protein biogenesis and translational control, by illustrating how multiple enzymatic activities are coordinated at the ribosomal exit site. The manuscript is clearly written and the figures well presented. We recommend publication after the following points are addressed.

We are happy about this reviewer's overall very positive evaluation.

General points:

While the structural and biochemical data convincingly reveal how NAA40 can be recruited to the ribosome and coordinated with NAC and METAP1, much of the evidence relies on *in vitro* reconstitution. The authors do point this out in the text (e.g. lines 254–257, 280–283), but it would be helpful if they more clearly emphasized the current lack of full *in vivo* confirmation and discussed how future work might address this.

We agree with the reviewer that the lack of *in vivo* confirmation should be clearly stressed, especially since all current data on multi-enzyme/-factor assembly like METAP1-NAA40, METAP1-NatA or METAP2-NatA heavily rely on *in vitro* data. Therefore, we expanded the second chapter

of the discussion where we already touch on the subject further (lines 305-309) highlighting future advances in *in situ* cryo-EM and in cell (sm)FRET.

Minor points

1. Please check whether referencing in the abstract complies with Nat. Comms. guidelines.

We thank the reviewer for the advice. To comply with Nat. Comms. formatting guidelines we removed all references from the abstract.

2. Line 49: write out the abbreviations “Ph” and “me” in full (or in parentheses) at first mention.

We wrote out the two abbreviations at first mention in the introduction in parentheses.

3. Fig. 1a: Clarify how histones, XRCC5/6, and PRKDC were identified (e.g. MS, immunoblotting) and provide supporting data. In addition, there is no negative control IP (e.g. FLAG-empty, untagged, or unrelated FLAG-protein) to assess specificity.

Histones, XRCC5/6 and PRKDC were identified by MS of SDS-PAGE gel cutouts of prominent bands observed after Coomassie staining. MS was performed and results analyzed by our in-house MS facility. This is now more clearly stated in the Methods section (lines 439-441) and indicated in the Figure legend. Additionally, a supplementary table with the MS results is attached as Supplementary Data Table 1. In the case presented in Fig. 1a, no negative control is necessary since no quantitative assessment has been made, simply an inventory of co-enriched proteins. Furthermore, (over-) expression of FLAG-tagged proteins followed by native co-IP has been performed multiple times from human cells with different bait proteins before throughout literature (e.g. Thoms M. *et al.* Science 2020; ; Huso V. L. *et al.* Nature 2025; Fiorentino F. *et al.* Nucleic Acids Research 2025; Ameismeier M. *et al.* Nature 2020; etc.) showing no unspecific co-purification of histones, XRCC5/6 or PRKDC, especially not at a Coomassie stainable level. Most importantly, this study is not focussing on expanding or completing the set of interactors for NAA40 but specifically on characterizing the co-translational activity of NAA40. Regardless, backed by MS and a recent publication that finds involvement of the yeast NAA40 homolog in DNA damage signaling (Klavaris A. *et al.* Epigenetics Chromatin 2025), we feel that co-enrichment of XRCC5/6 and PRKDC is noteworthy and might even provide the community with starting points for further investigations for the role of NAA40 in e.g. DNA damage or its nuclear role besides co-translational modification in the cytoplasm.

4. Line 88-90 and line 92 is referenced to Supplementary Fig. 1a and b, but should be 2a and b. In the text is written AF2, in the figure legend AF3.

We apologize for the mistake. Now, Supplementary Figures 3a-c and 3d are correctly referenced in lines 92-93 and line 95. We also corrected the Supplementary Figure legend to AF2 and 3 for either NAC-NAA40 or NAC.

5. Fig. 3a: there is a quite striking difference between lane 7 and 9. From the presented data, it seems likely, NAC Δ UBA occludes NAA40's weak direct binding to the ribosome. This could be tested by adding increasing amounts of NAC-deltaUBA to see if it displaces NAA40 from the ribosomes. Alternatively, titrating NAC vs. NAC-deltaUBA would strengthen the case that UBA-dependent binding is specific, not an artefact. Rephrase then the statement that NAC- Δ UBA "abolishes" ribosome binding accordingly. The assay is qualitative; additional controls would strengthen the data (e.g. NatA) or quantification or maybe mutants in the N-terminus of NAA40. Shift the legend in 3a slightly upward for clarity.

As suggested by the reviewer we performed an additional binding assay following our original experimental setup but in which we titrate NAC-wt vs NAC-deltaUBA. As expected, with increasing amounts of NAC-deltaUBA the co-precipitation/binding of ribosomes decreased until again no ribosomes were bound with NAC-deltaUBA alone (see new Supplementary Fig. 6e). Inversely, the amount of non-bound ribosomes in the flow-through fraction increases with the fraction of NAC-deltaUBA added (see new Supplementary Fig. 6f). We added a description to the Results section as requested (lines 163-170). While we cannot provide a definitive answer to this observation of apparent competitive binding, we speculate that the loss of ribosome binding of NAA40 with the NAC-deltaUBA might be caused by the flexible positioning of the NAC globular domain around the exit tunnel site leading to potential steric overlaps with NAA40 (more specifically its GNAT domain), and might therefore prohibit the interaction of (on its own weakly associated) NAA40 with the ribosome. This way, the UBA domain might provide an "allosteric effect" that mitigates the proper positioning of NAC on one hand and also enhances the binding of NAA40 on the other.

6. Antibody validation (Fig. 5e): Were the peptides biotinylated? It is nowhere stated in the methods and not clear. Was the antibody further purified to ensure Ac-binding, e.g. against free-Nt peptides binding and the FT against Ac-Nt binding?

We thank the reviewer for the comment. The peptides used for antibody generation and specificity testing were not biotinylated. The antibody was affinity-purified against peptides with and without N-terminal acetylation to ensure strict specificity for the N-terminally acetylated epitope. We also clarified that the peptides used for the Dot Blot Analysis were biotinylated (Supplementary Fig. 6g). This information has now been added in the revised Methods section and Figure legend.

7. Line 186: Residual Nt-acetylation in NACA-KD cells may also result from incomplete knockdown.

We thank the reviewer for this comment. qPCR analysis demonstrates that NACA transcript levels are reduced almost entirely in the knock-down cells (Supplementary Fig. 6i). While these data indicate that the knock-down is very effective, we cannot exclude the possibility that residual NACA protein or remaining NAC function contributes to the observed residual Nt-acetylation. We have clarified this point in the revised manuscript.

8. There is missing one H.S. in Supplementary Fig. 5h

We apologize for the oversight. H.S. was added to the Western blot annotation in Supplementary Fig. 6j.

Reviewer #2 (Remarks to the Author):

See replies to other reviewers. Thank you for your efforts in reviewing this manuscript.

Reviewer #3 (Remarks to the Author):

This study provides compelling structural evidence for NAC-mediated recruitment of NAA40 to ribosomes. Combined with biochemical data demonstrating that NACA knockdown reduces H4/H2A acetylation, these findings significantly advance our understanding of co-translational histone modification. Overall, this is an excellent study with high quality in experimental data and data presentation. However, the following major and minor issues require clarification in order to strengthen the manuscript.

We are happy about this reviewer's overall very positive evaluation.

Major Issues:

1. NAA40 binds both hibernating and active ribosomes, yet the cryo-EM structures show no engagement of its catalytic domain with the cognate substrate motif (Ser-Gly-Arg-Gly). This may arise from non-physiological NAA40 overexpression, permitting indiscriminate ribosome

association irrespective of nascent chain identity. To ensure substrate binding ability of NAA40., I recommend the authors to perform AlphaFold-Multimer predictions of NAA40 bound to H2A/H4 N-termini and check whether this binding mode conflicts with resolved NAA40-ribosome interactions.

We agree with the reviewer that overexpression can generally lead to increased indiscriminate ribosome association as becomes obvious with the large portion of NAA40-NAC bound hibernating ribosomes in our *ex vivo* cryo-EM data. However, in the manuscript we use the smaller subset of actively translating NAA40-NAC ribosomes to estimate the nascent chain length requirements for NAA40 mediated nascent histone acetylation (Fig. 1d). Since we cannot confidently assign density of the nascent chain within the ribosome or NAA40 to either of the (at least) two histone substrates we use a substrate analog bound crystal structure (Supplementary Fig. 3g) overlaid with our structure as reference. This crystal structure is essentially identical with another published crystal structure of N-terminal histone H2A/H4 peptide bound NAA40 (Magin R. S. *et al*, Structure 2015; PDB: 49UW). Since both substrate and substrate analog bound NAA40 structures were already determined experimentally at high resolution (which is in agreement with our structures) we don't think a computationally calculated structure model would add more insights. As illustrated by the dashed line in Fig. 1d connecting the nascent peptide density from the ribosome peptide tunnel exit to the NAA40 substrate cavity/catalytic site we do not observe any conflicts regarding substrate cavity accessibility when fitted to our cryo-EM data. To the contrary, as stated in the manuscript the cavity is facing towards the peptide exit site potentially allowing NAA40 to readily engage emerging substrates.

2. The NAA40 $\alpha 0$ helix mediates ribosome anchoring via rRNA (H19/H24/H46)—an interaction that is nascent chain-independent and topologically similar to NatA/E complexes. Given that NATs compete for overlapping ribosomal sites, I recommend the authors to discuss how NAC orchestrates histone-specific modification despite non-specific ribosome anchoring of NAA40.

The issue brought up by the reviewer is highly interesting but also highly complex and so far poorly understood. Indeed, some of the known nascent chain interacting/modifying factors and complexes share similar and/or overlapping binding sites on the ribosome, e. g. SRP and NAA40 or NatA/E, METAP2 and NAC, NMT1 and METAP1/2. Additionally, as ribosome binding is possible for any of these factors regardless of the presence of a nascent polypeptide, at least *in vitro* or under overexpression conditions in cells. As inferred by the reviewer, one would expect a degree of coordination for the different factors to be confidently directed to their target substrates within the competitive environment of the ribosome peptide exit site. Furthermore, not only is there an overlap of ribosome binding sites between NAA40 and NatA/E or SRP, but all of them benefit from

the same NACA UBA-domain interaction which adds an extra level of complexity. To entangle the coordination and interplay between these multiple factors and other not fully characterized ones like human NatB and NatC (especially in cells) will undoubtedly be one of the upcoming challenges for the field. To that end we can only speculate whether there is a constant exchange of competing factors on the ribosome until the correct factor and nascent chain match or whether there is indeed a more defined mechanism that relies on some sort of substrate pre-recognition by NAC that is then relayed by recruitment of the correct modifying enzyme. We prefer a hypothesis which would predict constant sampling of ribosomes of these factors and a change of off-rates dependent on substrate recognition. In any case, since we briefly pick up this topic in the first chapter of the discussion we now elaborate more on these possibilities and challenges there (lines 282-288).

Minor Issues:

1. Include pLDDT coloring and PAE metrics for all AlphaFold-predicted interactions (e.g., NAA40-NAC interfaces) to validate model confidence.

As requested we included pLDDT coloring and PAE matrices for all AlphaFold-predicted interactions.

2. "Horseradish peroxide" should be corrected to horseradish peroxidase (HRP) in Methods.

We corrected that mistake. The text now states: "horseradish peroxidase (HRP)".

Reviewer #4 (Remarks to the Author):

The paper provides novel structural insights into the interaction of the shortest Nat protein NatD to the ribosome. In contrast to other Nats this protein consists of a single subunit, NAA40, which specifically acetylates Histones H2A and H4 after their processing by MetAP1. NAA40 thereby is recruited close to the ribosome exit tunnel by interaction with NAC in a similar manner as shown previously for NatA and NatE. A critical role in recruitment seems to play the highly charged N-domain of NAA40 and the NACA Uba domain. Based on their structural data, the authors suggest that MetAP1 and NAA40 might be recruited concomitantly to the exit tunnel, possibly indicating a coordinated cleavage and acetylation reaction. The conclusions drawn from the structural data are supported by a variety of different biochemical experiments. Both, the structural data and the biochemical data, are of very high quality and seem to be consistent with the proposed

mechanisms. The manuscript is very well written and most of the Figures are very well prepared. Together with the importance of the NAA40 for its possible function in various tumors and our still limited understanding of the N-terminal modification reactions I strongly recommend this manuscript for publication in Nature Communications.

We are happy about this reviewer's overall very positive evaluation.

There are only some minor queries that should be addressed by the authors.

I would appreciate, if the authors could discuss the unique substrate specificity of NAA40 for H2A and H4 in some more detail.

This issue has been analyzed and explained in great detail in R.S. Magin, et al. Structure 2015, where the details of NAA40 substrate binding were investigated by X-ray crystallography and combined with mutational analysis. We added a short explanation to the introduction highlighting the difference in substrate recognition between Naa40 and other NATs with broader substrate range such as NatA and NatE (lines 45-47).

Given the copurification of XRCC5, XRCC6 and PRKDC a more global analysis of the extract shown in Fig. 1a by qMS might provide deeper and novel insights into the substrate specificity of NAA40. It would also be interesting if gammaH2A is present in the sample.

We agree with the reviewer that the co-purification of DNA damage response factors is intriguing as well as the question whether the substrate range of NAA40 extends from canonical histones H4/H2A. In the meantime, we have published a study demonstrating that H2A.X is a bona fide substrate of NAA40 (Klavaris A. *et al.* Epigenetics Chromatin 2025), further supporting a potential link between NAA40 and DNA damage related pathways (We have included this information in the Introduction (line 44) and updated paragraph 3 of the discussion (lines 321-323)). However, while we feel that the co-immunoprecipitation of XRCC5, XRCC6 and PRKDC is worthwhile mentioning in the manuscript to potentially provide initial evidence to further investigate the role of NAA40 in the nucleus and/or in DNA damage signaling (see Reviewer #1 Minor Point 3), it is not the focus of this study. Therefore, we believe that performing qMS to get deeper insights into NAA40 substrate specificity or interactors aside from NAC and the ribosome is beyond the scope of this study.

line 289 and 290: If the N-terminal helix of NAA40 is a major determinant for ribosome binding

and also harbors the nuclear localization signal, it is unlikely that the N-terminally truncated version serves in “potential nuclear post translational activity”. The authors should better explain what they mean here.

In the discussion portion mentioned by the reviewer only the canonical NAA40 is discussed but not the short form. However, as mentioned in the discussion, the cellular distribution of canonical NAA40 leaves certain questions for future research. For example, the full-length NAA40 is found mainly in the nucleus compared to the short version that is mainly located in the cytoplasm (PMID: 33916271 and compare Supplementary Fig. 6j: main and lower band for NAA40). This is counterintuitive since the $\alpha 0$ extension is functioning as main ribosome anchor as well as potential NLS, but the short version that lacks the N-terminal anchor and cannot associate with the ribosome is predominantly located in the cytoplasm. This distribution and its regulation, especially of full-length NAA40, might have a profound impact on its activity and would warrant further investigation in future work. Since the synthesis of histone proteins is for the most part tied to the cell cycle, one could for example imagine that the localization of NAA40 might also be linked to cell cycle progression.

So, when we mention in the results that NAA40S could take part in post-translational acetylation we still talk about activity within the cytoplasm (lines 183-185). To clarify this, we added “in the cytosol” to the sentence in question (now line 183-185).

Minor:

1: some of the Figures are too small to really see details and should be enlarged by the authors. This includes Figure 2a,3g, 4c, Extended Data Fig 1e, Extended Data Fig 2e, Extended Data Figure 3g and 3h and the Extended Data Figure 6b to 6e.

As requested, we enlarged the figure panels in question and adjusted the figures to accommodate the larger panels. For Supplementary Fig. 1e and 6b-e, we split the figures into one containing the cryo-EM sorting scheme and the other containing the reconstruction validation panels, each. These are now represented by Supplementary Fig. 1 and 2 or 7 and 8, respectively. Please note that splitting Supplementary Fig. 1 results in an incremental shift of +1 in numbering for all other Supplementary Figures thereafter that has been adopted within the text. We have enlarged all other panels in question to improve readability. We also want to note that lower resolution versions of the figures were provided to keep file sizes reasonable. The final figures will be provided as higher resolution versions which should also increase the readability.

2: On top of Figure 3a the legends table needs to be adjusted.

The table legend for Figure 3a has been adjusted and should now be better understandable.

3: Not sure what the authors mean with the question mark in Figure 3g. This should be removed or described in the legend.

Figure 3g shows an overlay of our NAA40-NAC assembly on the peptide tunnel exit site at the 60S ribosome overlaid with a surface representation of METAP1 taken from a previously reported structure. The idea behind the figure is to give the reader a picture of our hypothesis why concomitant binding of the three factors at the ribosome might be possible from a structural point of view instead of plainly describing the possibility in the text. We think this greatly helps readers - especially those who are not as familiar with structural data or previous publications on the topic - to visualize and understand a key point of why the following *in vitro* reconstitution experiments and cryo-EM analysis were conducted. We added the question mark for two reasons: 1) To indicate that we were questioning or wondering about the possibility of NAA40-NAC and METAP1 assembling together at the peptide exit. 2) While the panel serves as a visualization aid for our rationale we do not want readers to accidentally mistake the overlay shown for actual data. Therefore, we think it is justified to keep the question mark in the panel but we added an overall more detailed explanation to the figure legend as requested.

4: Figure 4a and b. I suppose the shown maps are filtered, which should be indicated in the text of the legend.

The figure panels show local resolution filtered densities. As requested, we added the description to the figure legend.

5: Figure 5: the colors of NACB and A are too pale and should be bit darkened to provide better contrast to the gray large subunit.

We adjusted the colors of all factors represented in the cartoon in Fig. 5 to give better contrast to the large ribosomal subunit.

6: Legend to Extended Data Figure 3: (line 797) f should be substituted by h

We corrected this mistake.

7: there happened some misassignment of Figures in the text: e.g. line 89/90 Extended Data Figure 1a actually refers to 2a: line 95 Extended Data Fig. 1a...;

We apologize and corrected the wrong assignment in formerly lines 89/90, Supplementary Fig. 3a-c is now correctly referenced. Formerly line 95 now correctly references Supplementary Fig. 1 and 3e,f. 1 being the cryo-EM sorting scheme showing how the subclass of actively translating 80S was derived and 3e,f showing overviews of the local resolution filtered map and the molecular

model as well as the density for the cut through views shown in main Fig. 1c where density has been replaced with surface representations of the molecular model for a clearer representation for the reader.

8: I suppose the sentence in lines 214/215 refers to the experimental data in Fig. 3 and not to the structural data as indicated.

We corrected that mistake.

9: Line 140: references missing for NatA/E

We added relevant references to the text passage in question.

Point-by-point response

REVIEWERS' COMMENTS

Reviewer #1 (Remarks to the Author):

The authors have satisfactorily addressed our concerns, and the manuscript is suitable for publication in Nature Communications.

We thank the reviewer for their positive assessment of our revised manuscript.

Reviewer #2 (Remarks to the Author):

We thank the reviewer for their efforts in co-reviewing this manuscript.

Reviewer #3 (Remarks to the Author):

The authors have addressed my concerns. Overall, this is a very important study and warrants the rapid publication.

We are glad to hear that we could address the reviewer's concerns in the revised manuscript.

Reviewer #4 (Remarks to the Author):

The manuscript significantly improved. I do not have further queries.

We are pleased to hear that we could improve the manuscript and address the reviewer's remarks.